# Sterically confined rearrangements of SARS-CoV-2 Spike protein control cell invasion

Esteban Dodero-Rojas[1], Jose N Onuchic[1,2,3,4]*, Paul Charles Whitford[5,6]*

[1]Center for Theoretical Biological Physics, Rice University, Houston, United States; [2]Department of Physics and Astronomy, Rice University, Houston, United States; [3]Department of Chemistry, Rice University, Houston, United States; [4]Department of Biosciences, Rice University, Houston, United States; [5]Center for Theoretical Biological Physics, Northeastern University, Boston, United States; [6]Department of Physics, Northeastern University, Boston, United States

**Abstract** Severe acute respiratory syndrome coronavirus 2 (SARS-CoV-2) is highly contagious, and transmission involves a series of processes that may be targeted by vaccines and therapeutics. During transmission, host cell invasion is controlled by a large-scale (200–300 Å) conformational change of the Spike protein. This conformational rearrangement leads to membrane fusion, which creates transmembrane pores through which the viral genome is passed to the host. During Spike-protein-mediated fusion, the fusion peptides must be released from the core of the protein and associate with the host membrane. While infection relies on this transition between the prefusion and postfusion conformations, there has yet to be a biophysical characterization reported for this rearrangement. That is, structures are available for the endpoints, though the intermediate conformational processes have not been described. Interestingly, the Spike protein possesses many post-translational modifications, in the form of branched glycans that flank the surface of the assembly. With the current lack of data on the pre-to-post transition, the precise role of glycans during cell invasion has also remained unclear. To provide an initial mechanistic description of the pre-to-post rearrangement, an all-atom model with simplified energetics was used to perform thousands of simulations in which the protein transitions between the prefusion and postfusion conformations. These simulations indicate that the steric composition of the glycans can induce a pause during the Spike protein conformational change. We additionally show that this glycan-induced delay provides a critical opportunity for the fusion peptides to capture the host cell. In contrast, in the absence of glycans, the viral particle would likely fail to enter the host. This analysis reveals how the glycosylation state can regulate infectivity, while providing a much-needed structural framework for studying the dynamics of this pervasive pathogen.

*For correspondence:
jonuchic@rice.edu (JNO);
p.whitford@northeastern.edu
(PCW)

Competing interests: The authors declare that no competing interests exist.

## Introduction

The current COVID-19 pandemic is being driven by severe acute respiratory syndrome coronavirus 2 (SARS-CoV-2). While vaccine and treatment development will help mitigate the immediate impact of this disease, long-term strategies for its eradication will rely on an understanding of the factors that control transmission. The need to isolate the molecular constituents that govern SARS-CoV-2 dynamics is widely recognized, where the global scientific community has undergone its most rapid transformation in recent history. This unprecedented redirection of scientific inquiry has rapidly provided atomic-resolution structures of SARS-CoV-2 proteins at various stages of infection (*Wrapp et al., 2020*; *Walls et al., 2020*; *Wang et al., 2020*; *Lan et al., 2020*; *Yuan et al., 2020*; *Watanabe et al., 2020a*), as well as computational analysis of specific conformational states (*Casalino et al., 2020*;

*Schlick et al., 2020*; *Roy et al., 2020*; *Ali and Vijayan, 2020*; *Verkhivker, 2020*; *Shin et al., 2020*; *Turoňová et al., 2020*). Despite these advances, our understanding of the mechanism by which SARS-CoV-2 enters the host cell is limited.

Central to the function of SARS-CoV-2 is host-cell recognition by the Spike protein, which results in virus-host membrane fusion and transfer of the viral genome. In the active virion, the Spike protein assembly (S) is a threefold symmetric homo-trimer (*Wrapp et al., 2020*), where each protein contains approximately 1200 residues (*Figure 1*). The complex is anchored to the viral membrane envelope by a transmembrane (TM) helical bundle, while the remaining regions reside on the exterior of the viral particle. Cleavage at the S1/S2 and S2' sites leads to activation of the Spike protein, where the resulting subunits (S1 and S2) maintain contact through non-bonded interactions (*Figure 1A*; *Shang et al., 2020*). The receptor binding domain (RBD) in S1 can then associate with the ACE2 receptor (*Letko et al., 2020*; *Lan et al., 2020*), which triggers S1 release from S2 (*Figure 1B*). While the order of S2' cleavage and S1/S2 dissociation is not known, it is generally thought that both

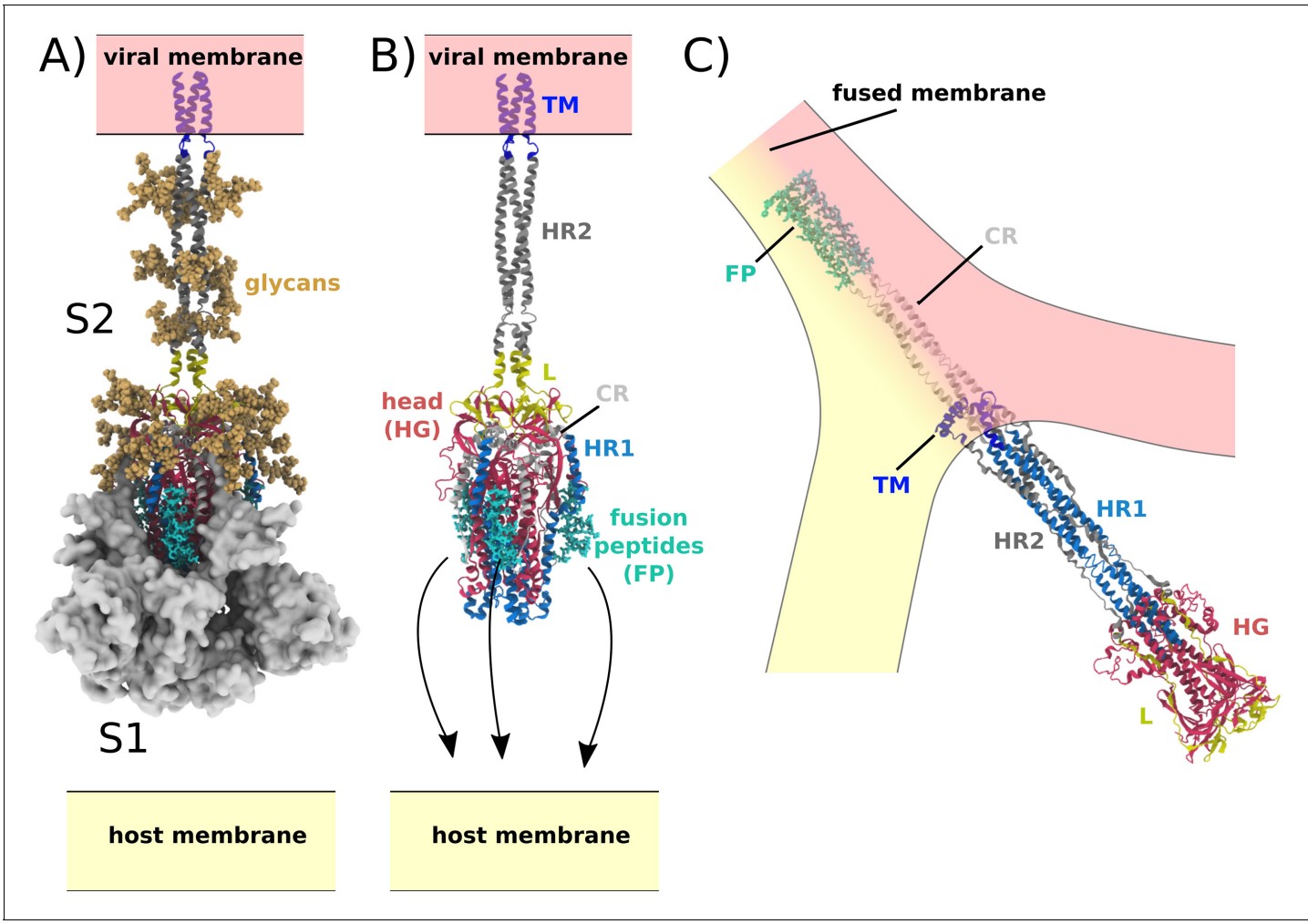

**Figure 1.** Spike-protein-mediated membrane fusion. (**A**) The active Spike protein assembly is composed of the subunits S1 (white surface) and S2 (cartoon representation) (*Walls et al., 2020*), which remain bound through nonbonded interactions. Numerous glycosylation sites (glycans shown in orange) are present in the Head Group (HG) and Heptad Repeat 2 (HR2) regions. The modeled glycans are consistent with previous studies (*Casalino et al., 2020*; *Turoňová et al., 2020*). (**B**) Upon recognition of the ACE2 receptor and cleavage at the S2' site, S1 dissociates. In addition to the HG and HR2 regions, S2 is composed of the Heptad Repeat 1 (HR1), Linker (L), Fusion Peptide (FP), Connector (CR), and Transmembrane (TM) regions. Since the HR2 and TM regions were not resolved in the prefusion structure (PDB ID: 6VXX [*Walls et al., 2020*]), they were modeled as a helical bundle, consistent with previous studies (*Casalino et al., 2020*). (**C**) Release of S1 allows for the FPs to associate with and recruit the host membrane. The HG and HR1 regions undergo a large-scale rotation (>90 degrees), which leads to fusion of the host and viral membranes. Since the CR and FP regions were not resolved in the postfusion structure (PDB ID: 6M3W [*Fan et al., 2020*]), they were modeled as an extended helical bundle.

processes occur prior to any large-scale rearrangements of S2. During the subsequent global structural rearrangement of S2, the fusion peptides must bind and recruit a host cell (*Figure 1C*; *Basso et al., 2016*).

Most of the current SARS-CoV-2 therapies and vaccines have focused on the ACE2 recognition stage of virus invasion. An alternate strategy is to target the conformational change in S2 that induces membrane fusion. In this direction, it has been found that disrupting formation of the HR1-HR2 six-helical bundle through introduction of inhibitor peptides can halt expression of the virus in biological samples (*Xia et al., 2020*). This suggests there is great potential for impeding viral entry by targeting intermediates of the Spike conformation change. Accordingly, understanding the conformational rearrangements associated with membrane fusion can provide new targets for medicinal applications. However, probing intermediate stages of fusion has proven to be extremely difficult. As a result, there has yet to be an experimental or computational/simulation study reported that provides direct insights into the mechanistic aspect of the S2 conformational change.

The fusion process involves global reorganization of S2 (*Figure 1* and *Appendix 1—figure 1*). This includes dissociation of the fusion peptides (FP) from the head group (HG), disordering of Heptad Repeat 2 (HR2), rotation of the head group relative to the viral membrane and then reordering of Heptad Repeat 1 (HR1), HR2 and Connecting Region (CR) into an extended helical arrangement. During this elaborate process, the fusion peptides associate with the host membrane, where subsequent 'zippering' of a HR1-HR2 superhelical structure likely provides energy to recruit the host membrane. In the postfusion structure, the TM, CR, and FP regions adopt proximal positions, allowing them to facilitate membrane fusion.

While high-resolution structures have been resolved for the Spike protein in the prefusion and postfusion conformations, the precise structural mechanism of fusion is unknown. As a consequence, the molecular factors that control this process have yet to be determined. For example, while many post-translational modifications (glycans) have been identified (13 and 9 on each S1 and S2 monomer)(*Watanabe et al., 2020a*; *Watanabe et al., 2020b*; *Walls et al., 2020*; *Wrapp et al., 2020*), there has not been an investigation into their role during the fusion step of infection. However, simulations of the prefusion protein have shown how glycans may shield the Spike protein and prevent recognition by the immune system (*Casalino et al., 2020*). In addition, studies have provided evidence that glycans may serve as activators for the lectin pathway (*Malaquias et al., 2021*; *Lenza et al., 2020*). While glycans have been implicated in other aspects of the viral 'life' cycle, it is not known whether they directly impact the host-entry process.

There are various challenges that have impeded the direct study of conformational changes in the S2 subunit. In terms of structural methods, due to the transient character of S2 intermediates, all previously reported structures are of the prefusion or postfusion states (*Wrapp et al., 2020*; *Walls et al., 2020*; *Fan et al., 2020*). While one could envision applying a range of simulation methods to gain insights into the dynamics of the transition, the size of the Spike protein makes many strategies intractable. For example, conventional explicit-solvent simulations can be used to study the detailed energetics of small proteins (*Lindorff-Larsen et al., 2011*). However, such highly detailed simulations of the full S2 trimer would be extremely computationally demanding, which would preclude the possibility of simulating the full conformational process. With this limitation, it can be advantageous to apply models that have simplified representations of the energetics, such as structure-based models. In a structure-based model, the potential energy function is defined based on knowledge of stable (i.e. experimental) conformations (*Clementi et al., 2000*). In the cases of protein folding and functional dynamics, these models have been able to provide descriptions of mechanisms that are consistent with experimental measures for various systems. The success of these simplified models to capture folding dynamics is a reflection of the strong limitations that are imposed by molecular sterics and the complexity of folded conformations (*Shea et al., 1999*; *Gosavi et al., 2006*). Inspired by studies of folding, all-atom variants were later used to simulate rearrangements in large assemblies, such as ribosomes (*Nguyen and Whitford, 2016*; *Levi et al., 2020*) or viral capsids (*Noel et al., 2016*; *Whitford et al., 2020*). Despite the simplicity of the models, mechanistic aspects of the dynamics are often robust to the model parameters. Similar to the study of folding, robustness can be understood as arising from the atomic resolution of the models, where steric interactions strongly limit the possible mechanistic properties (*Levi et al., 2019*). Motivated by the observation of sterics-associated robustness during folding and functional processes,

we will adopt this modeling strategy to study the dynamics of the S2 subunit as it transitions between the prefusion and postfusion conformations.

Here, we performed molecular dynamics simulations with an all-atom structure-based model to determine whether the steric composition of glycans can have a meaningful influence on SARS-CoV-2 Spike-protein-mediated membrane fusion. Simulations were initiated with the Spike protein in the prefusion conformation, while the energetics were defined to favor the postfusion conformation (shown in *Figure 1C*). It is important to emphasize that the prefusion model used as a starting point is intended to represent a state in which S2' cleavage and S1 dissociation have occurred. While the precise timing of these steps is unknown, we assume they can occur prior to any significant conformational changes in the S2 subunit. By comparing the dynamics with and without glycans present, we show how the steric composition of the glycans can extend the lifetime of a critical intermediate in which the head appears to become sterically-caged. This leads to a transient pause that may increase the probability of recruiting the host cell. These calculations provide physical evidence that the glycosylation state is a critical factor that determines infectivity of SARS-CoV-2.

## Results

### Simulating the membrane-fusion-associated conformational change of SARS-CoV-2 Spike protein

In order to characterize the mechanism of Spike-protein-mediated membrane fusion, we employed an all-atom structure-based model (*Whitford et al., 2009*; *Noel et al., 2016*) and simulated transitions between the prefusion and postfusion conformations (*Figure 2B* and *Video 1*). In a structure-based (Gō-like) model, some/all of the energetic interactions are defined based on knowledge of specific stable (experimentally resolved) structures. In the context of protein folding, applying these types of models (*Clementi et al., 2000*) is supported by the principle of minimal frustration (*Bryngelson and Wolynes, 1989*; *Bryngelson et al., 1995*). However, to warrant their application to study conformational transitions, it is necessary to recognize that the models describe the effective energetics of each system (*Hyeon and Thirumalai, 2011*; *Di Pierro et al., 2018*; *Chan et al., 2011*). That is, by explicitly defining the molecular interactions to stabilize the endpoint conformations, the models are intended to provide a first-order approximation to the energetics. In the presented model, only interactions that are specific to the prefusion and postfusion configurations were defined to be stable. For the TM region, stabilizing prefusion-specific interactions allow it to serve as an anchor between the Spike protein and the viral membrane. An implicit membrane potential was also introduced to restrain the TM to a plane. Even though an orientation bias was not introduced, the TM region generally remains nearly perpendicular to the viral membrane surface (*Appendix 1— figure 2*). Finally, all non-TM interactions were defined to stabilize the postfusion conformation (*Figure 1*). Qualitatively, this model describes the Spike protein as a loaded (non-linear) spring that is released upon cleavage of the S2' site and dissociation of S1 (*Figure 2A*). While the potential energy in the model is downhill, molecular sterics can still lead to pronounced free-energy barriers that control the kinetics (*Whitford and Onuchic, 2015*; *Levi et al., 2019*).

Before describing the simulated events, it is valuable to discuss the analysis of energetic frustration (*Ferreiro et al., 2007*; *Parra et al., 2016*) in the Spike protein, which supports the application of an unfrustrated model. Frustration analysis of the postfusion structure indicates there is a low degree of frustration in HR1, HG, L and most of HR2 (*Appendix 1—figure 3*). In contrast, for the prefusion conformation, there is a higher degree of frustration in HR1 and HR2. This suggests that HR1 and HR2 are only marginally stable in the prefusion conformation after S2' cleavage and S1 dissociation. In the structure-based model, interactions with these regions are defined to only stabilize the postfusion conformation, which is consistent with the reduced degree of frustration in the postfusion state. While this analysis supports the use of an unfrustrated model to study dynamics, we find that FP, CR and the C-terminal end of HR2 are frustrated in the postfusion conformation. For HR2, the higher level of frustration in the C-terminal region may be understood in terms of its local environment. That is, frustration analysis is based on the energetics of proteins in solvent. However, the C-terminal end of HR2 is positioned adjacent to the viral membrane, which likely introduces interactions that can stabilize its structure. This interpretation is supported by frustration analysis of other proteins, where highly-frustrated regions have been found to frequently engage in binding

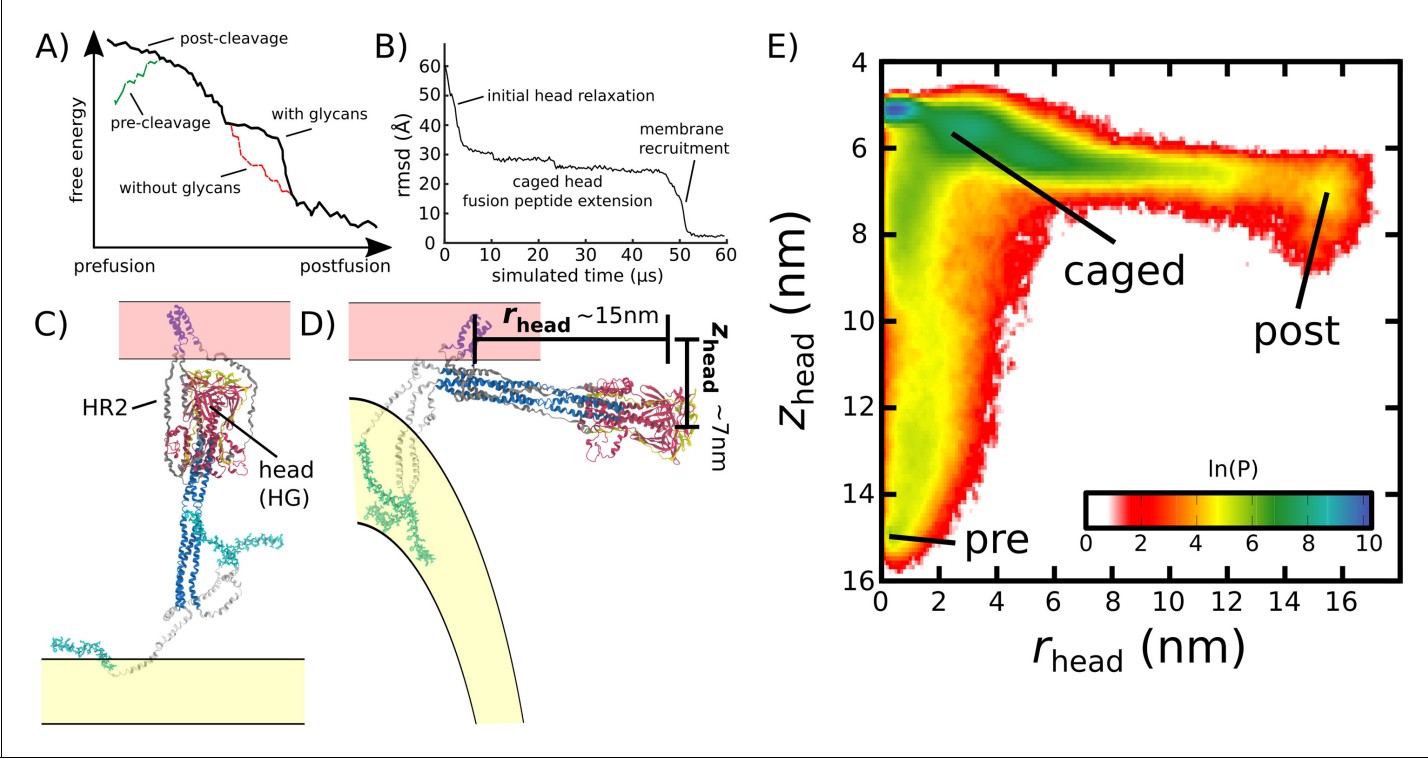

**Figure 2.** Simulating Spike-protein-mediated membrane fusion. Simulations with an all-atom structure-based model (*Whitford et al., 2009*; *Noel et al., 2016*) allow for transitions between prefusion and postfusion configurations to be observed. (**A**) Schematic representation of the energetics in the structure-based model. The postfusion configuration was defined as the global potential energy minimum. The pre-cleavage state (green) is assumed to be stable, where cleavage and release of S1 leads to an unstable prefusion configuration (black). While, in the employed model, stabilizing energetic terms favor the postfusion configuration, steric interactions between the protein and glycans may impede the motion (black vs. red). (**B**) Representative simulation (1 of 1000) of the pre-to-post transition. Spatial rmsd from the post configuration (excluding TM, CR, and FPs) is shown, as a function of time. The simulation included explicit glycans, as well as an effective viral membrane potential. (**C**) After an initial relaxation phase (panel B), the head (red) appears to become caged by the HR2 strands (gray), allowing it to sample configurations near the viral membrane (pink). While the host membrane (yellow) was not included in the simulations, it is depicted for illustrative purposes. (**D**) After reaching the caged ensemble, the head escapes and the HR1-HR2 superhelix assembles. The position of the head group, relative to the TM region (blue), is described by the cylindrical coordinates $r_{head}$ and $z_{head}$. The origin is defined as the geometric center of TM, and the cylindrical axis is perpendicular to the viral membrane. See Materials and methods for details. (**E**) Probability distribution of simulated events (with glycans) reveals an obligatory cage-like intermediate.

interactions (*Ferreiro et al., 2007*). Consistent with this, FP engages in host membrane interactions, again consistent with an elevated level of energetic frustration (*Gorgun et al., 2021*). Similarly, due to its proximity to FP, CR is also likely to engage in membrane interactions, which is suggested by the elevated level of predicted frustration. As a final note, for the postfusion conformation, CR and FP were modeled based on helical template structures, where imperfections in the structures may contribute to higher levels of predicted frustration. In summary, the majority of the postfusion structure is predicted to be minimally frustrated, which supports the use of a structure-based model. Further, as discussed in subsequent sections, the current study is focused on the dynamics of structure formation in HR1, HG, L, and HR2, whereas CR and FP remain largely disordered in the simulated events. Accordingly, any frustration in these regions that was not included in the model is not likely to influence the primary finding of the current study.

To investigate the dynamics of Spike-protein-mediated membrane fusion, we simulated thousands of transitions between the prefusion and postfusion conformations of S2 (*Video 1*). To describe the rearrangements, we considered the distance between HG and the viral membrane ($z_{head}$), as well as the displacement of HG parallel to the membrane ($r_{head}$; *Figure 2D*). The probability distribution as a function of $r_{head}$ and $z_{head}$ (*Figure 2E*) shows a clear ordering of HG rearrangements. Each simulation was initialized in the prefusion conformation ($r_{head} = 0$, $z_{head} \approx 15$ nm). From there, the head moves towards the viral membrane (i.e. decreasing values of $z_{head}$), and the HR2

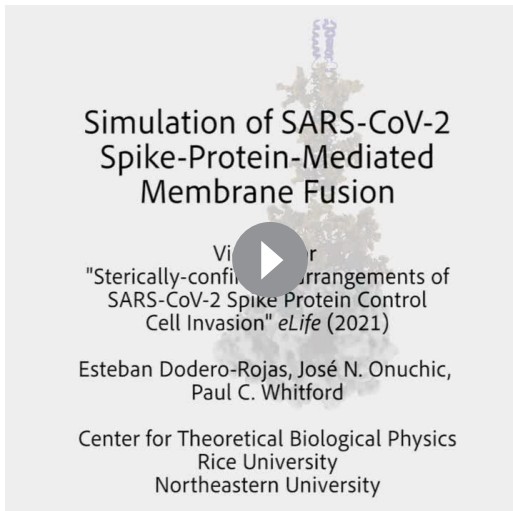

**Video 1.** This video shows a representative simulation (1 of 1000) of the fully-glycosylated S2 subunit of the SARS-CoV-2 protein as it transitions from the prefusion to the postfusion configuration.

https://elifesciences.org/articles/70362#video1

strands appear to enclose HG. In this 'caged' ensemble, the long axis of HG remains roughly perpendicular to the membrane (*Appendix 1—figure 4D*). During initial relaxation of HG, the fusion peptides simultaneously extend toward the host membrane (*Figure 2C*). After relaxation of HG, it then rotates away from its vertical orientation (increasing values of $r_{head}$; *Figure 2D*) by passing between two of the HR2 strands. As the head rotates (*Appendix 1—figure 4E-G*), the FP and CR regions are drawn toward the viral membrane. The simulations were terminated when all non-CR and non-FP residues adopted their postfusion orientations. Since the CR and FP regions were not resolved in the postfusion structure (*Fan et al., 2020*), the simulations describe formation of all experimentally resolved structural elements.

In simulations, the ordering of conformational events is robust to the presence of glycans. When glycans were explicitly included, there were only minor differences in the range of HG configurations that are sampled (*Figure 2E* vs. *Appendix 1—figure 5*). In both cases, HG initially relaxes towards the viral membrane before rotating towards the host as shown in *Figure 2* and *Appendix 1—figure 5*.

## Glycans induce a long-lived sterically caged intermediate

We find that glycans can reduce the kinetics of HG rearrangements by introducing a dynamic steric cage that confines HG to a position near the viral membrane (*Figure 3A*). This caging process gives rise to prolonged sampling of an intermediate (*Figure 2E*; $z_{\text{head}} \leq 6$ nm, $r_{\text{head}} \leq 4$ nm) in which the long axis of HG is roughly perpendicular to the viral membrane (*Appendix 1—figure 4*). The lifetime of the caged intermediate is given by $\tau_{\text{cage}} = \tau_{\text{exit}} - \tau_{\text{enter}}$ (*Figure 3B*). $\tau_{\text{enter}}$ is defined as the time at which the assembly enters the intermediate (i.e. when $z_{head}$ first decreases below 6.5 nm). $\tau_{\text{exit}}$ is the time at which $r_{head}$ first exceeds 5 nm, indicating the head has been displaced outside of the cage-like formation (*Figure 2D*). For the representative trajectory show in *Figure 3B*, $\tau_{\text{cage}}$ is roughly 37 μs. See Materials and methods for estimation of time units in this model.

The lifetime of the caged intermediate strongly depends on the presence of glycans. For the glycan-free system, $\tau_{\text{cage}}$ values were narrowly distributed, where $\bar{\tau}_{\text{cage}} = 6.7 \mu s$ (*Figure 3D*). When glycans are present, the distribution has a tail that extends to much larger values (100–500 μs; *Figure 3C*), and $\bar{\tau}_{\text{cage}}$ increases nearly 5-fold (29.7 μs). To isolate the origins of this effect, we repeated our simulations with subsets of glycans present. In one set of simulations, only glycans on HG were included, while the other set only included glycans on HR2 (*Appendix 1—figure 6*). Interestingly, the HG-glycan model exhibited timescales that were comparable to those obtained for the fully glycosylated system. In addition, the HR2-glycan model yielded timescales that were comparable to those obtained when S2 is not glycosylated (*Appendix 1—figure 6*). These comparisons reveal that the glycan-associated increase in excluded volume of HG is a dominant factor that determines the kinetics of interconversion between prefusion and postfusion conformations. Finally, glycans are likely to exhibit a degree of attraction under cellular conditions, which may lead to slower intramolecular diffusion. Accordingly, the predicted glycan-induced reduction in kinetics should represent a lower-bound on the influence of glycosylation.

It is important to emphasize that the apparent glycan-dependent kinetics may be fully attributed to steric effects. That is, while the protein energetics were explicitly defined to favor the postfusion conformation, glycans were not assigned energetically-preferred conformations. Rather, the potential energy of the glycans only ensured that stereochemistry and excluded volume were preserved.

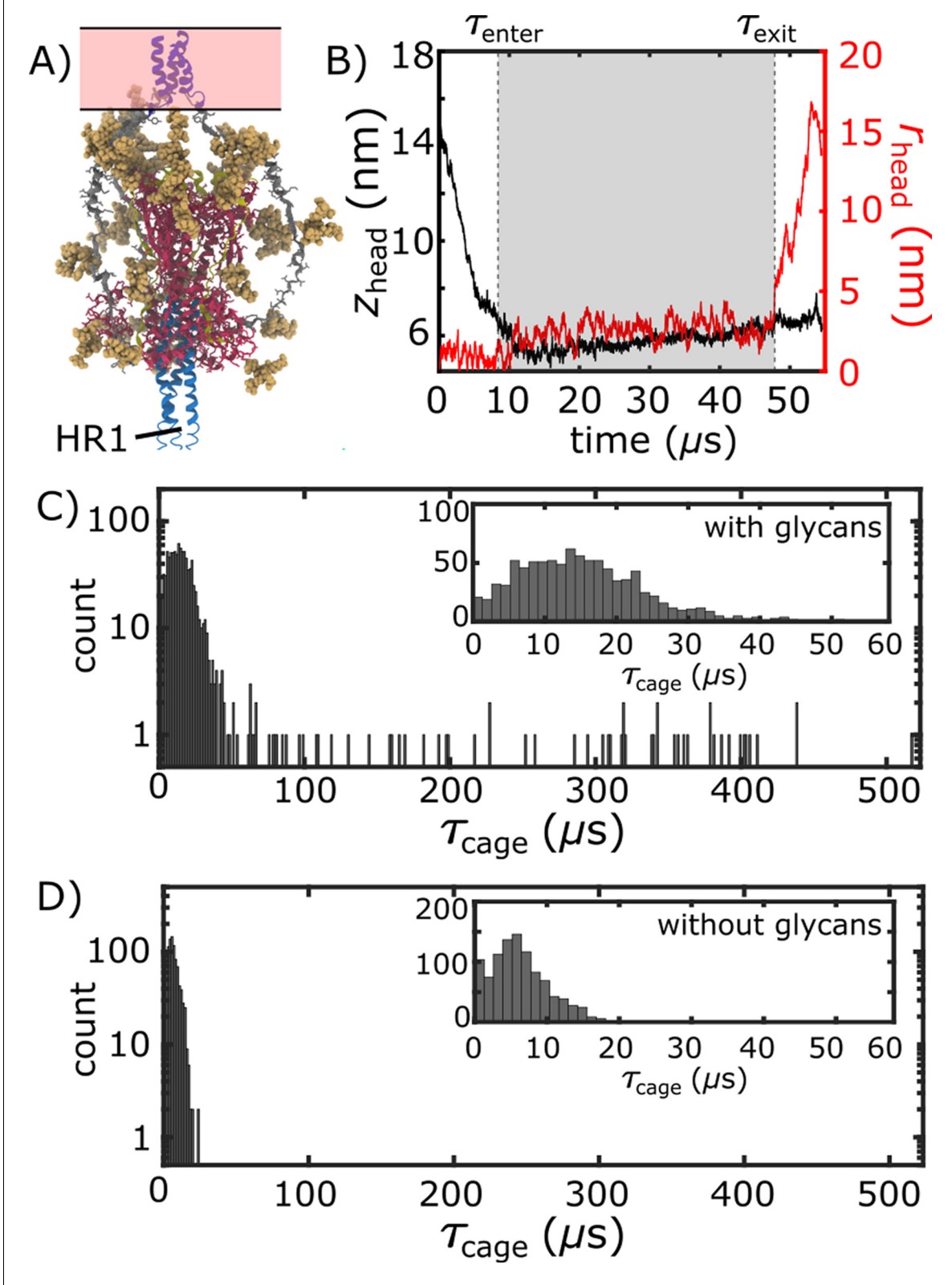

**Figure 3.** Glycan-induced caging of the head domain. Glycans impede head rearrangement by introducing a steric cage. (A) Snapshot from the caged ensemble illustrates the high density of glycans surrounding the head. (B) To define the duration of each caging event ($\tau_{cage} = \tau_{exit} - \tau_{enter}$), we measured $z_{head}$ and $r_{head}$ (*Figure 2D*). Based on the 2D probability distribution (*Figure 2E*), the system was defined as entering the cage when $z_{head}$ first drops below 6.5 nm: $\tau_{enter}$. $\tau_{exit}$ is the time at which the head moves laterally, relative to the trans-membrane region ($r_{head} > 5$ nm). (C) Distribution

*Figure 3 continued*

of $\tau_{\text{cage}}$ values when glycans are present. There is an extended tail at large values ($100 - 500 \mu s$). (**D**) When glycans are absent, the $\tau_{\text{cage}}$ values are narrowly distributed around short timescales.

In addition, the excluded volume interactions were purely repulsive. Thus, the observed reduction in rate for HG motion is due solely to the excluded volume of the glycans, and not the formation of stabilizing interactions. In terms of modeling considerations, the ability to attribute this effect entirely to steric interactions means that the effect will be present in simulations with any atomic-resolution model. While the precise kinetic properties of the system will depend on the details of each model, the influence of glycan sterics that is predicted by the structure-based model will be robust to the exact energetic representation that is applied.

## Glycan cage promotes host membrane capture

The simulated trajectories suggest that glycan-associated attenuation of head rearrangements can facilitate host membrane recruitment and fusion. As described above, we find that the steric composition of the glycans introduces a highly crowded environment, which can transiently cage the HG domain (*Figure 3A*). We additionally find that initial relaxation of HG is rapid (*Figure 3B*) where caging introduces a pause that allows the HR1, FP and CR regions to sample extended configurations. To describe structure formation of the HR1 region, we calculated the fraction of postfusion-specific contacts that are formed as a function of time, $Q_{\text{HR1}}$. Calculating the number of 'native' contacts formed is motivated by protein folding studies, which have shown it to be a reliable measure of structure formation (*Cho et al., 2006*). When glycans are absent, HG frequently exits the cage prior to reaching the postfusion structure of HR1 ($Q_{\text{HR1}} = 1200 - 1300$; *Figure 4B*). In contrast, when glycans are present, HR1 is typically fully-formed ($Q_{\text{HR1}} > 1400$) before HG exits the steric cage (*Figure 4A*). By caging HG in a position that is perpendicular to the viral membrane (*Figure 3A*), glycans help to ensure that the newly assembled HR1 helical coil remains directed towards the host membrane. This orientation of HR1 may serve to facilitate host membrane capture by the FP and CR regions.

To quantify the likelihood that the Spike protein will associate with and recruit the host, we considered the extension of each fusion peptide from the viral membrane. For this, we first defined a putative host-membrane distance, $d_{host}$, which was set to discrete values (22–38 nm). Again, while the host membrane was not included in these simulations, this host-membrane distance is an indicator of the region that needs to be visited by the FP to successfully bind the host. We then determined whether the distance between the viral membrane and FP ($d_{\text{FP}}$, *Figure 5A*) exceeded $d_{host}$ for each of the three FP tails in the assembly (*Figure 5B*). $P_{\text{capture}}$ was then defined as the probability that at least one FP extends to the host membrane (*Figure 5C*). We use the notation $P_{\text{capture}}$, since one expects that the extension of the FPs will be correlated with the probability that the Spike protein successfully captures the host cell.

We find that glycosylation of S2 significantly increases the probability that a FP will extend to the host membrane (*Figure 5C*). We further partitioned the capture distributions by calculating the probability that exactly 1, 2, or 3 FPs cross the host membrane (*Figure 5D–F*). Interestingly, we find that all distributions exhibit significant differences for $26 < d_{\text{host}} < 34$ nm. In the absence of glycans, the probability of associating 3 FPs is nearly 0 in this range, while the probability increases to ~0.1 when glycans are present (*Figure 5F*). Similarly, the probability that exactly 2 FPs will cross the host membrane is ~0 for $d_{\text{host}} > 28$ nm, when glycans are absent. The probability then increases to ~0.1 when glycans are present (*Figure 5E*). Finally, for $26 < d_{\text{host}} < 33$, the probability that one FP will cross the host membrane is increased by ~0.2 when glycans are present (*Figure 5D*). To ask whether this differential extension of the FP is due to tilting of TM, we calculated the distribution of angles between the TM bundle and the viral membrane (*Appendix 1—figure 2*). This revealed that the TM tilting distributions were similar for the two systems, which indicates that the differential dynamics of the FPs can not be attributed to this aspect of the model. Finally, while the Spike Protein has been experimentally observed to spontaneously transition from the prefusion to postfusion configuration (*Cai et al., 2020*), the probabilities reported above describe events that occur when the Spike Protein is activated through host-virus interactions.

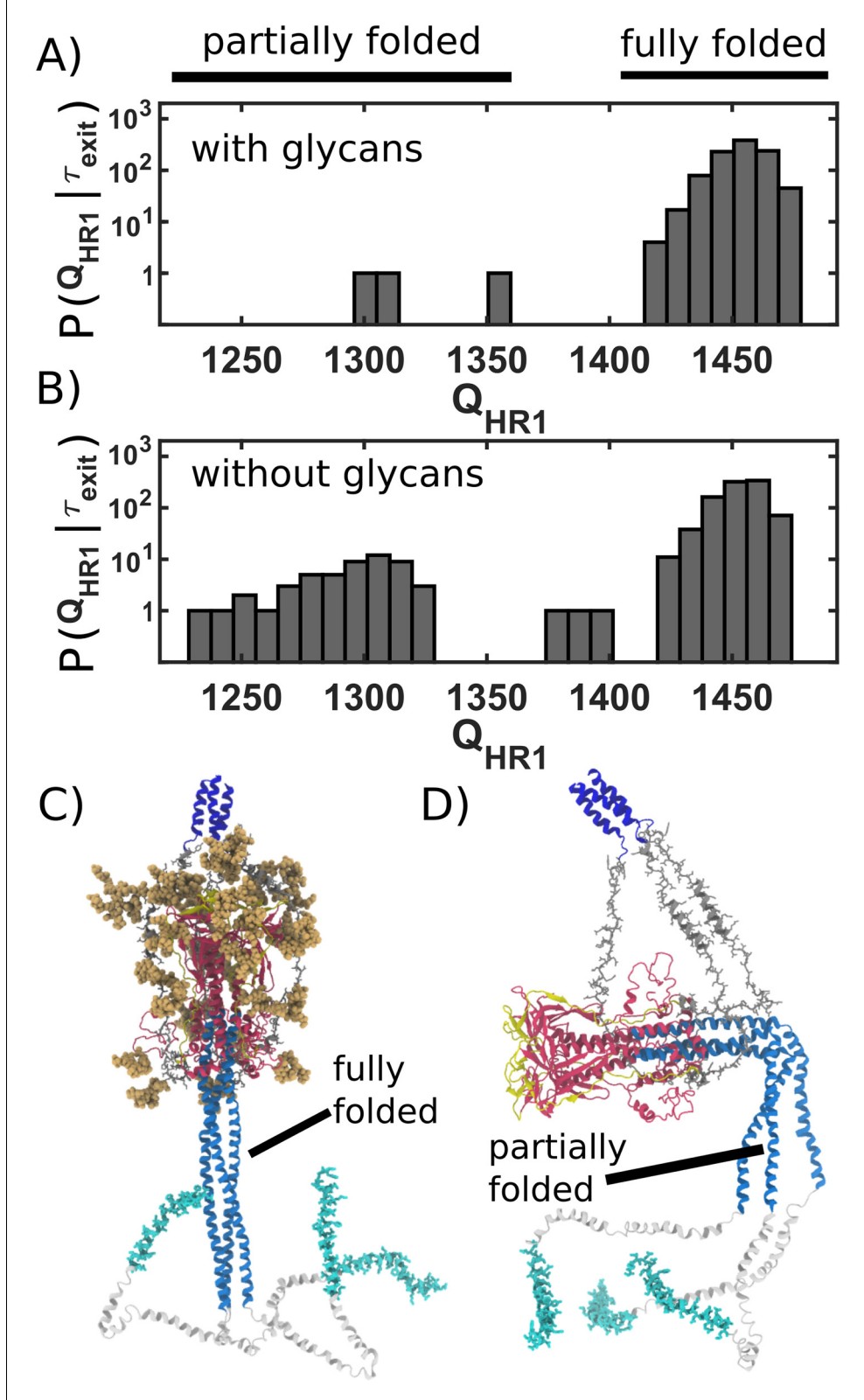

**Figure 4.** Caging of head allows for extension of HR1 helix. (**A**) Distribution of $Q_{HR1}$ (number of postfusion-specific HR1 contacts) values when glycans are present. Distribution describes the first frame in each simulation for which the head is outside of the steric cage. In all but three simulations, nearly all HR1 contacts (>1420 of 1489) are formed upon exit of the cage. (**B**) Distribution when glycans are not included. When glycans are absent, it is common that HR1 is not completely formed (i.e. $Q_{HR1}$ < 1350) prior to HG escape. (**C**) Representative snapshot of a caged structure in which HR1 is fully

*Figure 4 continued on next page*

*Figure 4 continued*

formed and extended toward the host. (D) Representative snapshot of a glycan-free case where the head escapes prior to fully forming HR1. As a result, HR1 can adopt bent configurations.

The glycan-dependent probability of host-membrane association suggests several features of the fusion process. Cryotomography imaging has revealed that the virus-host inter-membrane distance is approximately 30 nm during infection (*Turoňová et al., 2020*). Based on this, our simulations indicate that, if the Spike protein were not glycosylated, it is most likely that none of the FPs would associate with the host. Since Spike protein rearrangements are irreversible, these failed attempts would represent lost opportunities to infect the cell. Therefore, these simulations suggest the probability of infection would drop substantially if the Spike were not glycosylated. Consistent with this, experimental measurements have found that inhibiting the production of glycans decreases the efficiency of host cell entry (*Yang et al., 2020*). To assess whether the influence of glycans on FP dynamics is robust, we considered a variant of our model in which the TM helices may dissociate from each other. Specifically, the intra-TM harmonic interactions were replaced with weaker 6–12 Lennard-Jones interactions (*Appendix 1—figure 7*). With this modified-TM potential, we simulated 1000 transitions for the glycan-present and glycan-absent systems (2000 events, in total). These simulations show that the differential dynamics of the FPs is robust to the precise description of the TM bundle.

When glycans are absent, there is a marginal probability that only one or two FPs will reach the membrane (3D-E), where the other FPs would likely transition directly to their postfusion orientations without engaging the host. Such a process has been described as a 'cooperative' mechanism in fusion proteins class I, such as Hemagglutinin A, where only a fraction of the FPs anchor to the host membrane, while the remaining FPs bind to the viral membrane (*Lin et al., 2014*). However, when glycans are present, there is a non-negligible probability that three FPs will reach the host membrane. When all three FPs attach to the host membrane, the dynamics may be described in terms of the so-called 'sequential' mechanism of fusion (*Lin et al., 2014*). Together, these observations demonstrate how the steric contribution of glycans is critical to the mechanism and likelihood of cell invasion by SARS-CoV-2.

In terms of the mechanistic features of membrane fusion, one can expect that FP binding to a host membrane will impact the probability that subsequent FPs will also bind. To explore this point, we introduced a second flat-bottom potential (Appendix Equation 1) to describe the host membrane. Using this extended model, we simulated 1000 pre-to-post transitions for the glycosylated Spike protein. We find that the probability of unsuccessful activation (i.e. 0 FPs captured) is not affected by the host potential (*Appendix 1—figure 8*). However, introducing the effective host membrane potential significantly increases the probability that all three fusion peptides reach the host. This demonstrates how anchoring the first FP helps the Spike protein maintain an orientation that favors additional FP binding events. Together, this analysis suggests that the primary mode for Spike protein mediated membrane fusion is through use of a 'sequential' mechanism.

## Discussion

The ongoing COVID-19 pandemic requires the rapid identification of molecular factors that enable infection. A necessary step during infection involves virus/cell membrane fusion, which is mediated by a major conformational change of the Spike protein. Here, we propose a mechanism where, after cleavage and dissociation of S1, sufficient time has to be made available for the fusion peptides to reach the cell membrane, before the conformational change in S2 can complete. Using all-atom models with simplified energetics, we have shown how the steric composition of post-translational modifications may introduce the delay necessary for such a mechanism to be utilized. This glycan-induced pause appears to allow for an extended window during which the fusion peptides may search for the host cell (*Figure 6*). In simulations that did not include glycans, the Spike protein was most likely to adopt the postfusion configuration without extending the fusion peptides towards the host. Thus, in the glycan-free case, the protein can bypass the sterically-caged intermediate, leading to failed attempts to capture the host cell. These findings suggest an interesting theoretical

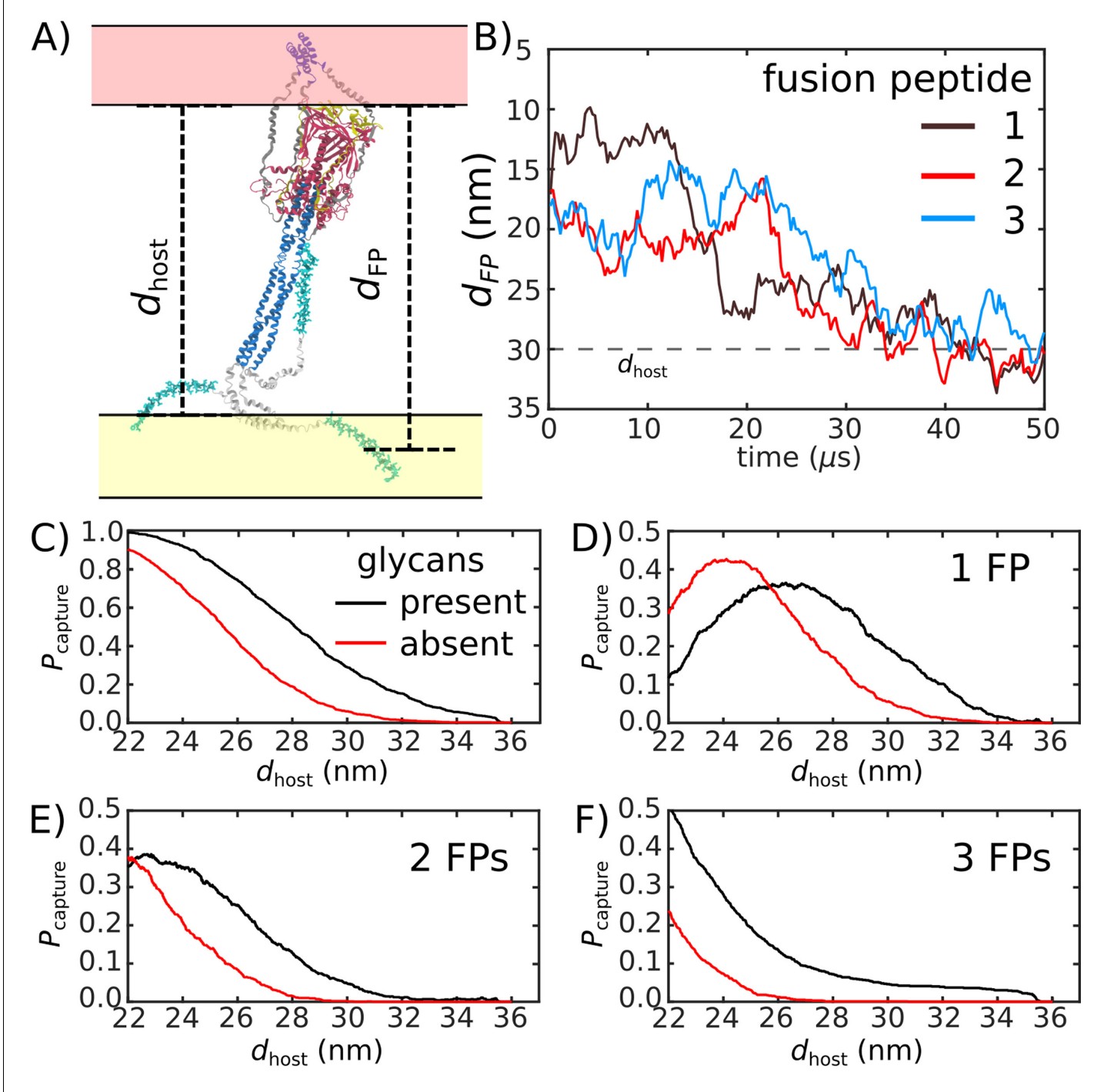

**Figure 5.** Glycans promote host capture. (A) Snapshot of the glycosylated Spike protein with the head domain in a caged configuration (glycans not shown). Caging allows the fusion peptide tails to extend toward and engage the host membrane. $d_{FP}$ is the distance of the center of mass of each fusion peptide from the viral membrane surface. To calculate the probability of host capture, different values of the virion-host distance ($d_{host}$) were considered. (B) Representative simulated trajectory, showing $d_{FP}$ for each of the fusion peptides in a single S2 subunit. For reference, a $d_{host}$ value of 30 nm is indicated by a dashed line. (C) The probability of membrane fusion is expected to be proportional to the probability that at least one tail extends to the host membrane ($d_{FP}>d_{host}$). There is a higher probability of extending to larger $d_{host}$ values when glycans are present (black vs. red curves). This is due to the glycan-induced delay of head rotation (**Figure 2**), which ensures the HR1 helix remains directed towards the host as the FPs sample extended configurations. (D–F) The probability that 1, 2, or 3 FPs exceed $d_{host}$. In all cases, the presence of glycans shifts the distribution to larger values of $d_{host}$, indicating an increased probability of capturing the host. This reveals a critical role for glycans during cell invasion.

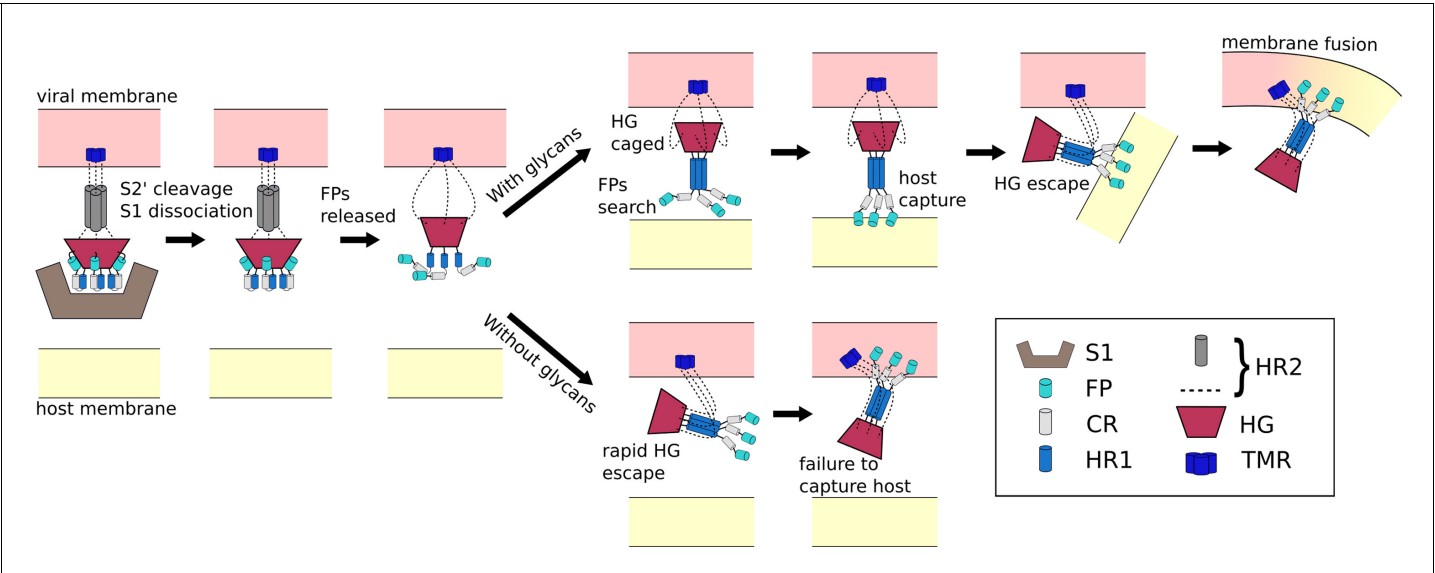

**Figure 6.** Schematic of fusion mechanism of the Spike protein. Initial activation of the Spike protein (left) is associated with release of S1, which is triggered by cleavage at the S2' site and ACE2 receptor binding. When glycans are present (top), HG will enter a caged ensemble where the FPs search for, and capture, the host membrane. HG can then escape the cage, which draws the viral and host membranes together and leads to fusion. In the absence of glycans (bottom), HG can bypass the caged ensemble, resulting in a failed attempt to recruit the host.

prediction that the precise glycan composition is a critical factor that determines transmissibility of SARS-CoV-2.

The current predictions suggest that the steric composition of the Spike protein and glycans can guide the global dynamics of host-membrane capture. While these results are intuitive, it is possible that non-specific stabilizing interactions (i.e. not found in the pre, or post, fusion conformations) can have a notable influence on the rearrangement. For example, there may be specific long-lived non-native interactions that can transiently maintain the orientation of the HG region and facilitate FP capture of the host. Another limitation of the current analysis is that our model describes interactions between the Spike and effective membrane regions in terms of a short-range effect. If long-range electrostatic interactions dominate the FP-membrane association process, then it is possible that an alternate sequence of events may be observed. Finally, another avenue for further study would be to consider so-called 'multi-basin' structure-based models (*Whitford et al., 2007*). In such approaches, each element of the Spike protein would have interactions that stabilize both the pre and postfusion conformations. This would allow one to identify the influence of competing stabilizing interactions that may impede the 'downhill' dynamics of the current model. For example, introducing prefusion contacts in the HR2 region would likely delay entry of HG into the cage. Similarly, prefusion interactions could extend the time required for the FPs to initially dissociate from S2. With these open questions in mind, it will be interesting to see the extent to which various factors can enhance or attenuate the steric signatures that are described in the current study. In this context, the presented simulations provide a foundation for understanding and quantifying the relative contributions of each biophysical factor during this large-scale motion.

With the range of possible contributors to Spike protein dynamics, there are clear opportunities for novel experiments to reveal the precise influence of glycosylation on Spike protein mediated cell entry. To test the predictions of the current study, one may consider applying site-specific mutations, in order to inhibit glycan binding at individual residues. These altered Spike protiens could then be integrated into pseudovirus particles (*Yang et al., 2020*), which would allow one to measure the impact of specific glycans on the ability of the Spike to associate with a cell. If sterics dominate the dynamics, as predicted by the structure-based model, then mutations to HG glycan sites should significantly reduce the probability of membrane capture.

In addition to providing immediate insights into the influence of glycans on Spike protein dynamics, the current simulations establish a foundation for experimentally and theoretically investigating

other factors that may influence cell invasion. To give one example, the presented models may be extended to account for electrostatic and solvation effects. With ongoing advances in high-performance computing, combined with the relatively low computational cost associated with these models, many variations may be explored in the coming months that will help elucidate the full range of factors that control this deadly pathogen.

## Materials and methods

### Structural modeling of the spike protein

Since complete structures of the full-length SARS-CoV-2 Spike protein have not been resolved experimentally, for either the prefusion or postfusion states, structural modeling steps were applied prior to performing simulations. For this, we used a cryo-EM structure of the prefusion assembly (PDB ID: 6VXX [*Walls et al., 2020*]), which lacks residues 828–853 and 1148–1233 (found in HG, and the HR2 and TM regions). For the postfusion state, we used a structural model of the SARS-CoV-1 system (PDB ID: 6M3W [*Fan et al., 2020*]) as a template for constructing a homology model of the SARS-CoV-2 system. However, the postfusion model was lacking residues 772–918 and 1197–1233 (FP, CR and TM). In addition, since available structures only partially resolved the base of each glycan, we constructed structural models of the complete glycans for both states (pre and post). For the prefusion structure, models of the TM and HR2 regions were constructed using the homology modeling webtool of SWISS-MODEL (*Waterhouse et al., 2018*). Consistent with the study of *Casalino et al., 2020*, both regions were modeled as coiled coils, where the sequence was assigned Uniprot (*UniProt Consortium, 2019*) sequence P0DTC2-1. This was accomplished with the automodel module of Modeller 9.24 (*Sali and Blundell, 1993*), with restraints included to preserve symmetry. The resulting model was threefold symmetric, where the RMSD between monomers was less than 1 Å. For the postfusion structure, unresolved residues in FP and CR were modeled as helical regions, using the automodel module of Modeller 9.24 with symmetry restraints imposed. CR and FP were modeled as coiled coils connected by short disordered loops. Homology models were constructed based on the structure of a coiled coil template (PDB ID: 2WPQ [*Hartmann et al., 2009*]). The TM strands were assigned alpha helical structures using Modeller 9.24. As a note, the postfusion configuration of the TM region was not used to define any aspect of the structure-based model.

Glycans were added to both structural models using the Glycan Reader Charmm server (*Park et al., 2019*). The same glycan composition was used as in other recent studies (*Casalino et al., 2020*; *Watanabe et al., 2020a*). A complete list of modeled glycans can be found in *Appendix table 1*. The glycosylated structural models of the prefusion and postfusion systems that were generated in this study are provided in the Supplementary Material.

### All-atom structure-based model

All simulations employed an all-atom structure-based model to describe the Spike protein, with additional restraints imposed on the TM region, as well as an effective viral membrane potential. To describe the energetics of the protein, a structure-based model was constructed based on the postfusion model, using the default parameters in SMOG 2 (described in *Noel et al., 2016*). Several modifications were introduced to the force field, as described below. Non-default parameters were assigned for bond lengths and angles, as well as planar dihedrals. Rather than using the values found in the cryo-EM structure, bond lengths and angles were given the values found in the Amber03 force field (*Duan et al., 2003*). The strengths of non-planar dihedrals and contacts were consistent with earlier implementations of the structure-based model (*Noel et al., 2016*), these interactions are further described in the SI (Appendix Equation 2). Contacts were identified using the Shadow algorithm (*Noel et al., 2012*). A complete description of this variant of the model is described in *Whitford et al., 2020*. Force field definition files, which include glycans, are available for download at https://smog-server.org (SMOG2 force field repository ID: AA_glycans_Dodero21.v1). To ensure that the TM region remains in a helical bundle arrangement, contacts in the TM region were replaced by harmonic interactions, with distances taken from the prefusion conformation. To mimic the presence of a viral membrane, a flat-bottom potential shown in Appendix Equation 1 was imposed on the TM region to limit the movement to be inside the putative membrane region. Also, to avoid non-TM residues from crossing the effective membrane, a repulsive inverted harmonic flat

bottom potential, beginning at the putative position of the viral membrane surface, was applied to atoms in HG. The potential was set to 0 at $z_{\text{head}} = 2$ and the harmonic constant was set to two reduced energy units per $nm^2$.

## MD simulations

All simulations were performed using the GROMACS software package (v2020.2) (*Lindahl et al., 2001*; *Hess et al., 2008*) with source code modifications to implement the Gaussian-based flat bottom potential (Appendix Equation 1 and *Appendix 1—figure 9*). Input files for Gromacs were generated using the SMOG 2 software package (*Noel et al., 2016*), while additional in-house scripts were used to subsequently modify the force field. Simulations of seven different systems were performed: glycan-free, fully-glycosylated, HR2-glycosylated, HG-glycosylated, glycan-free with anharmonic TM interactions, glycans present with anharmonic TM interactions and glycans present with an effective host membrane potential. A total of 1000 transitions between the prefusion and postfusion structures were simulated for each system/model (7000 simulations, in total). Each system was first energy minimized using steepest descent energy minimization. Simulations were then performed using Langevin Dynamics protocols, with a reduced temperature of 0.58 (70 Gromacs units). In preliminary simulations, it was found that the assembly begins to unfold at a temperature of around 0.8. A timestep of 0.002 was used, and each simulation was continued until $r_{\text{head}}$ reached a value greater than 8 nm, which indicated that HG had escaped from the HR1 cage. To estimate the effective simulated timescale, we use the conversion factor of 1 reduced unit being equivalent to 1 ns (*Yang et al., 2019*), which was previously obtained based on the comparison of diffusion coefficients in a SMOG model and explicit-solvent simulations.

## Structural metrics

The following coordinates were used to describe the global rearrangement of the Spike protein:

- $z_{head}$ : To calculate $z_{head}$, the vector between the centers of mass of TM (residues 1203–1233) and HG (residues 1033–1129) was calculated and then decomposed into components that are perpendicular and parallel to the membrane plane. $z_{head}$ is the component that is perpendicular to the plane.
- $r_{head}$ : To calculate $r_{head}$, the vector between the centers of mass of TM (residues 1203–1233) and HG (residues 1033–1129) was calculated and then decomposed into components that are perpendicular and parallel to the membrane plane. $r_{head}$ is the component that is parallel to the plane.
- θ : Angle formed between the first principal axis of HG (residues 1033–1129) and the vector normal to the membrane. A value of 0 indicates that the HG is perpendicular to the viral membrane.
- $Q_{\text{HR1}}$: Number of postfusion-specific contacts formed (within 1.2 times the distance in the post-fusion conformation).

## Frustration analysis

The frustration analysis was performed using the Frustratometer Web Server (*Parra et al., 2016*). The analysis reports the degree of frustration around each residue. For two atoms to be defined as a contact, their distance must be less than 5Å. The degree of frustration is calculated based on energetic profiles obtained with the AWSEM force field (*Davtyan et al., 2012*).

## Acknowledgements

Work at the Center for Theoretical Biological Physics was supported by the NSF (Grant PHY-2019745). JNO was supported by the National Science Foundation (NSF) grants CHE-1614101 and PHY-1522550 and by the Welch Foundation (Grant C-1792). PCW was supported by NSF grant MCB-1915843. JNO is a Cancer Prevention Research in Texas Scholar in Cancer Research. We also acknowledge generous support from the Northeastern University Discovery cluster and Northeastern University Research Computing staff. Also, we are grateful for generous computational resources and support provided by the AMD COVID-19 HPC Fund program.

# Additional information

## Funding

| Funder | Grant reference number | Author |
|---|---|---|
| National Science Foundation | CHE-1614101 | Jose N Onuchic |
| National Science Foundation | PHY-1522550 | Jose N Onuchic |
| Welch Foundation | C-1792 | Jose N Onuchic |
| National Science Foundation | MCB-1915843 | Paul Charles Whitford |
| National Science Foundation | PHY-2019745 | Jose N Onuchic |
| Cancer Prevention and Research Institute of Texas | | Jose N Onuchic |

The funders had no role in study design, data collection and interpretation, or the decision to submit the work for publication.

## Author contributions

Esteban Dodero-Rojas, Conceptualization, Formal analysis, Methodology, Writing - original draft, Writing - review and editing; Jose N Onuchic, Conceptualization, Supervision, Methodology, Writing - review and editing; Paul Charles Whitford, Conceptualization, Formal analysis, Supervision, Methodology, Writing - original draft, Writing - review and editing

## Author ORCIDs

Esteban Dodero-Rojas https://orcid.org/0000-0001-9054-9555
Jose N Onuchic https://orcid.org/0000-0002-9448-0388
Paul Charles Whitford https://orcid.org/0000-0001-7104-2265

## Decision letter and Author response

Decision letter https://doi.org/10.7554/eLife.70362.sa1
Author response https://doi.org/10.7554/eLife.70362.sa2

# Additional files

## Supplementary files

- Supplementary file 1. Structural Model of Prefusion Structure.
- Supplementary file 2. Structural Model of Postfusion Structure.
- Transparent reporting form

## Data availability

The current manuscript is a computational study. Simulations were prepared with SMOG 2 (free, open-source), which is available at smog-server.org. Force field templates for SMOG 2 are available for download through the SMOG 2 Force Field Repository on the smog-server page. Simulations were performed using Gromacs (free, open-source).

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

# Appendix 1

## Results

### TM tilt angle distributions

In the virtual membrane model used in this study, the TM region is restrained to a 5-nm-thick region, centered about the x-y plane. Since the TM is free to move along the plane, it is possible that the TM region can exhibit transient tilting, with respect the z direction. To address the potential effect of TM tilt on the overall character of the pre-to-post transition, we defined the tilt in terms of the angle formed by the z-axis and the axis of the TM. The TM axis is defined as the vector connecting the center of mass of the Cα atoms of residues 1213 (one in each chain) and the center of mass of the the Cα atoms of residues 1233. Then we computed the distribution of tilt angles for the glycan-present and glycan-absent simulations (*Appendix 1—figure 2*).

We found that the tilt angle distributions for both sets of simulations are similar, indicating that the effect of the TM tilt will be comparable for the glycosylated or deglycosylated systems. Accordingly, TM tilting does not contribute to the observed differential extension of the FPs. In addition, it is important to note that the tilt angle is less than 40° for more than 90% of the sample configurations. Taking into account the length of the TM region (~2.6 nm), a tilt of 40° would only account for a retraction of the Spike by ~0.6 nm. As apparent in *Figure 5* of the main text, independent of the effect of tilting, the simulated Spike protein would still reach physiological relevant distances (~30 nm).

### Effect of a virtual host membrane on FP capture probabilities

It may be expected that after one FP binds to the host membrane, the other FPs will be more likely to bind as well. In order to quantify this effect, we implemented a virtual host membrane potential that can capture each FP. The host membrane was modeled as a flat-bottom potential (Appendix *Equation S1*) of width ~5 nm. Since recent computational studies (*Gorgun et al., 2021*) suggest that host membrane binding is irreversible, we set the depth of the potential acting on each atom to 4 $k_BT$. Since this is stronger than the accessible thermal energy, the simulated capture events were observed to occur irreversibly.

After introducing this potential, we simulated 1000 pre-to-post transitions of the glycosylated Spike protein. As a control, we first verified that the inclusion of the host membrane potential did not affect the probability that the Spike will misfire (i.e. 0 FPs captured). As shown in *Appendix 1— figure 8*, the probability that the Spike fails to capture the host membrane is comparable when the membrane potential is absent or present. With regards to the number of FPs that are captured in successful events, we find that the presence of the virtual host membrane substantially increases the number of times that 3 FPs reach the host.

## Methods

Membrane flat-bottom potential

$$V(z) = \begin{cases} K\left[1 - \exp\left(\frac{-\left(z - \frac{w_m}{2}\right)^2}{2\sigma^2}\right)\right], & \frac{w_m}{2} < z \\ 0, & -\frac{w_m}{2} < z < \frac{w_m}{2} \\ K\left[1 - \exp\left(\frac{-\left(z + \frac{w_m}{2}\right)^2}{2\sigma^2}\right)\right], & z < -\frac{w_m}{2} \end{cases} \tag{S1}$$

Here, $z$ is the distance of the center of mass of TM from the center of the effective membrane. $w_m$ represents the membrane width (5 nm), and the depth of the potential $K$ was set to 2 reduced energy units (4 $k_BT$), while $\sigma = 0.2$ nm.

Single-basin structure-based model potential

$$V = \sum_{bonds} \frac{\epsilon_r}{2}\left(r_{ij} - r_{ij,0}\right)^2 + \sum_{angles} \frac{\epsilon_\theta}{2}\left(\theta_{ijk} - \theta_{ijk,0}\right)^2 +$$

$$+ \sum_{impropers} \frac{\epsilon_{\chi_{imp}}}{2}\left(\chi_{ijkl} - \chi_{ijkl,0}\right)^2 + \sum_{planar} \frac{\epsilon_{\chi_{planar}}}{2} F_P\left(\varphi_{ijkl}\right) +$$

$$+ \sum_{backbone} \epsilon_{bb} F_D\left(\phi_{ijkl} - \phi_{ijkl,0}\right) + \sum_{sidechains} \epsilon_{sc} F_D\left(\phi_{ijkl} - \phi_{ijkl,0}\right) +$$

$$+ \sum_{contacts} \epsilon_C \left[\left(\frac{\sigma_{ij}}{r_{ij}}\right)^{12} - 2\left(\frac{\sigma_{ij}}{r_{ij}}\right)^6\right]$$

$$+ \sum_{non-contacts} \epsilon_{nc} \left(\frac{\sigma_{nc}}{r_{ij}}\right)^{12}. \tag{S2}$$

Here, we define $F_D(\phi) = [1 - \cos(\phi)] + \frac{1}{2}[1 - \cos(3\phi)]$, as used in earlier implementations of structure-based models (**Noel et al., 2016**), and $F_P(\varphi) = [1 - \cos(2\varphi)]$. The bonded parameters, such as bond lengths and angles ($r_{ij,0}$ and $\theta_{ijk,0}$), were obtained from the AMBER03 force field (**Duan et al., 2003**). The planar dihedrals were maintained by cosine potentials of periodicity 2. All non-planar dihedral angles were assigned 1–3 periodicity cosine dihedral potentials ($F_D$) with minima corresponding to the postfusion conformation $\phi_{ijkl,0}$. Stabilizing 6–12 Lennard-Jones interactions were introduced for all atom pairs that are in contact in the postfusion conformation, with minima set to the distances found in the postfusion conformation $\sigma_{ij}$.

## Generated structural models

The structural models of the complete glycosylated systems are included as SI documents.

prefusion model The prefusion model is included in the file prefusion_Spike_DoderoRojas.pdb.

postfusion model The postfusion model is included in the file postfusion_Spike_DoderoRojas.pdb.

**Appendix 1—table 1.** N-glycan listing.
Complete list of N-glycans included in the simulations.

| | |
|---|---|
| N706 | aDMan(1→6)[aDMan(1→3)]aDMan(1→6)[aDMan(1→3)]bDMan(1→4)bDGlcNAc(1→4)bDGlcNAc(1→) |
| N717 | aDMan(1→6)[aDMan(1→3)]aDMan(1→6)[aDMan(1→2)aDMan(1→3)]bDMan(1→4)bDGlcNAc(1→4)bDGlcNAc(1→) |
| N801 | aDMan(1→6)[aDMan(1→3)]aDMan(1→6)[aDMan(1→3)]bDMan(1→4)bDGlcNAc(1→4)bDGlcNAc(1→) |
| N1074 | aDMan(1→6)[aDMan(1→3)]aDMan(1→6)[aDMan(1→3)]bDMan(1→4)bDGlcNAc(1→4)bDGlcNAc(1→) |
| N1098 | aDNeu5Ac(2→6)bDGal(1→4)bDGlcNAc(1→2)aDMan(1→3)[aDMan(1→6)[aDMan(1→3)]aDMan(1→6)]bDMan(1→4)bDGlcNAc(1→4)bDGlcNAc(1→) |
| N1134 | bDGlcNAc(1→2)aDMan(1→6)[bDGlcNAc(1→2)aDMan(1→3)]bDMan(1→4)bDGlcNAc(1→4)[aLFuc(1→6)]bDGlcNAc(1→) |
| N1158 | bDGlcNAc(1→2)aDMan(1→6)[bDGlcNAc(1→2)aDMan(1→3)]bDMan(1→4)bDGlcNAc(1→4)bDGlcNAc(1→) |
| N1173 | bDGlcNAc(1→6)[bDGlcNAc(1→2)]aDMan(1→6)[bDGlcNAc(1→4)[bDGlcNAc(1→2)]aDMan(1→3)]bDMan(1→4)bDGlcNAc(1→4)[aLFuc(1→6)]bDGlcNAc(1→) |
| N1198 | aDNeu5Ac(2→6)bDGal(1→4)bDGlcNAc(1→6)[bDGal(1→4)bDGlcNAc(1→2)]aDMan(1→6)[bDGal(1→4)bDGlcNAc(1→4)[bDGal(1→4)bDGlcNAc(1→2)]aDMan(1→3)]bDMan(1→4)bDGlcNAc(1→4)[aLFuc(1→6)]bDGlcNAc(1→) |

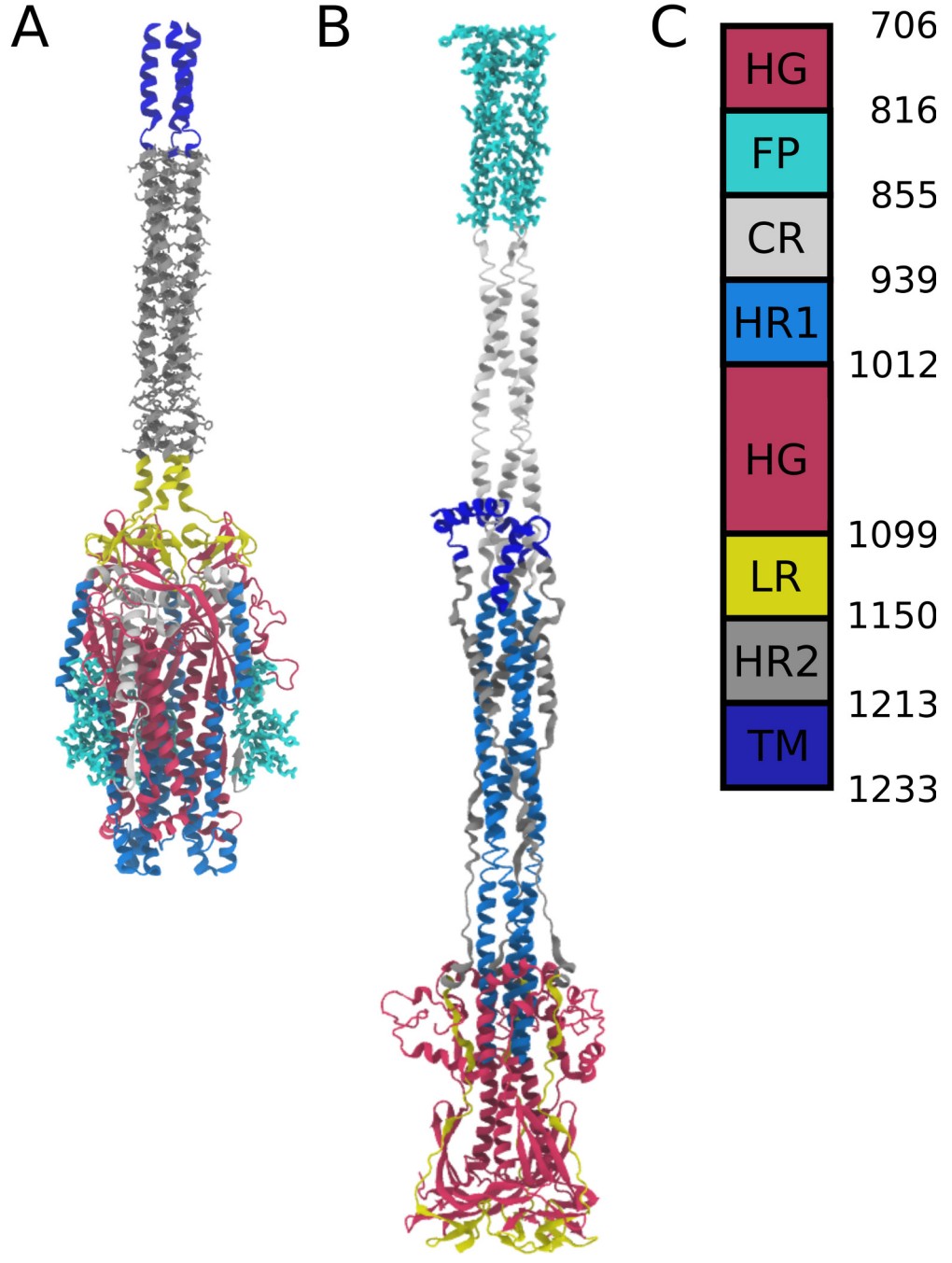

**Appendix 1—figure 1.** Definitions of domains within the S2 protein. (**A**) Prefusion S2 subunit structure of the Spike protein. (**B**) Postfusion S2 subunit structure of the Spike protein. (**C**) Sequence range of the Head Group (HG), Fusion Peptide (FP), Connecting Region (CR), Heptad Repeat 1 (HR1), Linker Region (LR), Heptad Repeat 2 (HR2), and Transmembrane Region (TM).

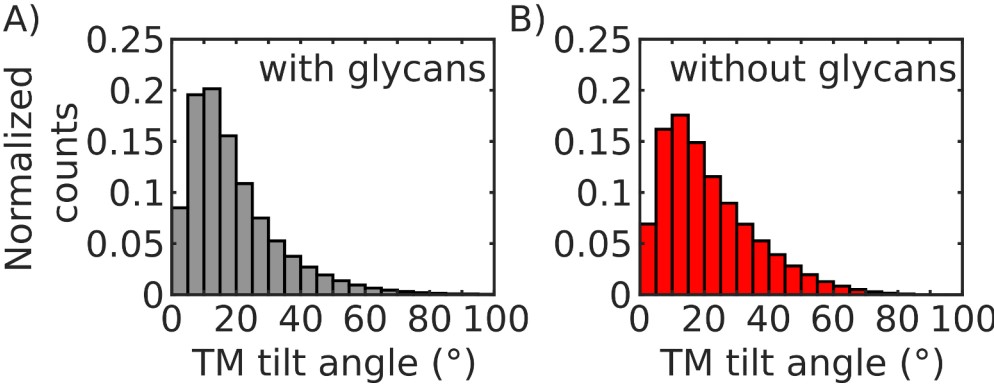

**Appendix 1—figure 2.** TM tilt angle distributions. (**A**) Distribution of TM tilt angles (defined in SI results section 1.1) sampled during simulations when glycans are present. (**B**) Distribution of TM tilt angles sampled during simulations when glycans are absent.

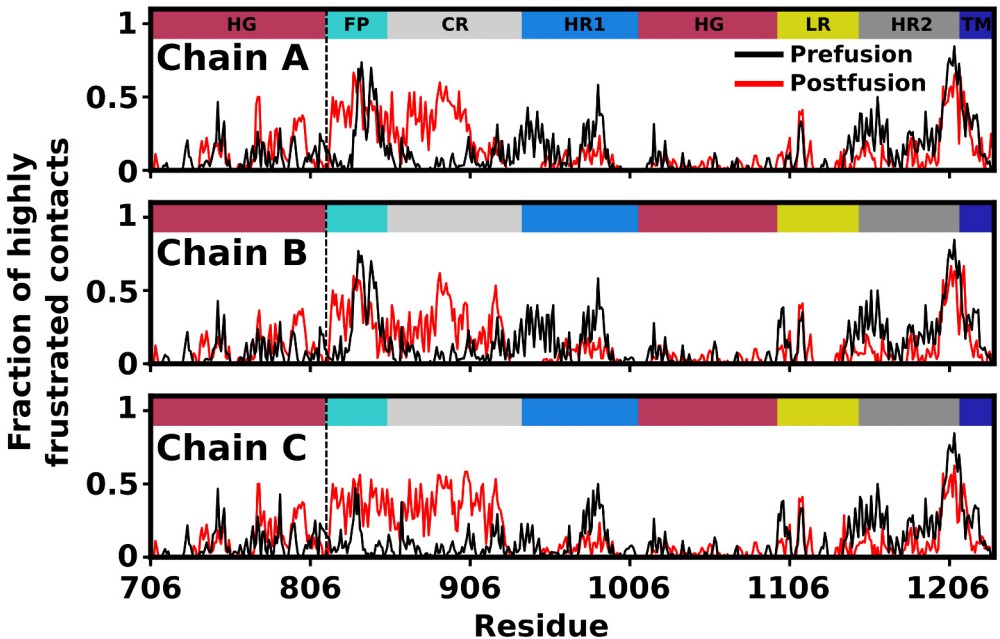

**Appendix 1—figure 3.** Predicted degree of frustration, by residue. Density of highly frustrated contacts in a 5Å sphere per residue for prefusion (black) and postfusion (red) S2 subunit structures. Dashed line represents the S2' cleavage site.

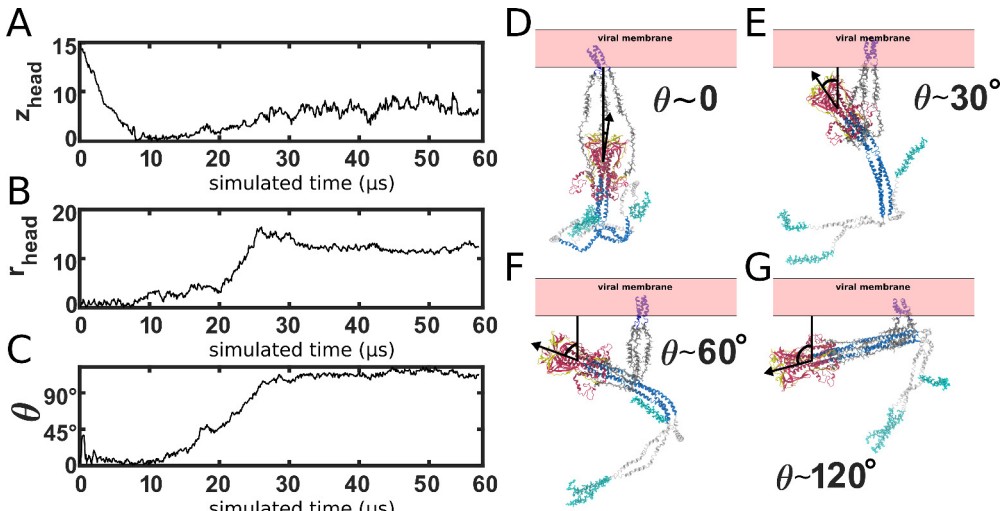

**Appendix 1—figure 4.** HG rotation. (**A-C**) Single time trace of $z_{head}$, $r_{head}$ and the HG principal axis polar angle, θ. (**D–G**) Snapshots of the orientation of HG, relative to the membrane. During the prefusion-to-postfusion transition, the head rotates from an orientation in which it is pointing toward the membrane, to an orientation where it is pointing away. Structural snapshots illustrate various orientations during the transition.

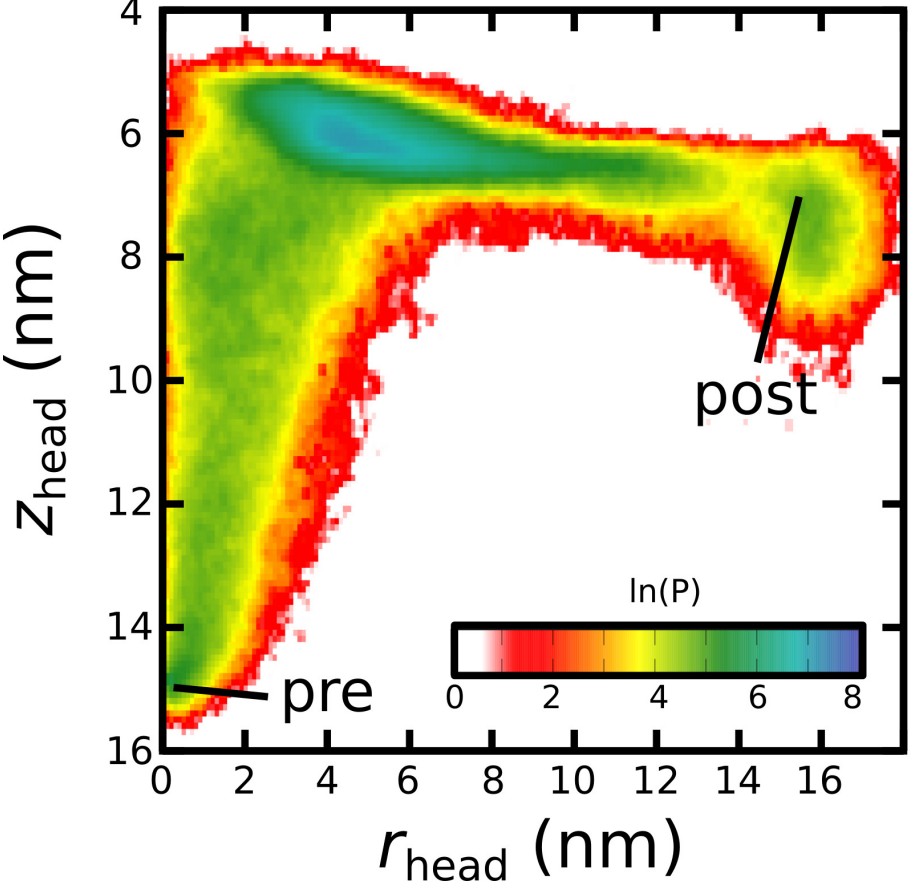

*Appendix 1—figure 5 continued*

**Appendix 1—figure 5.** Probability distribution when glycans are absent. Distribution calculated from 1000 independent simulations without glycans.

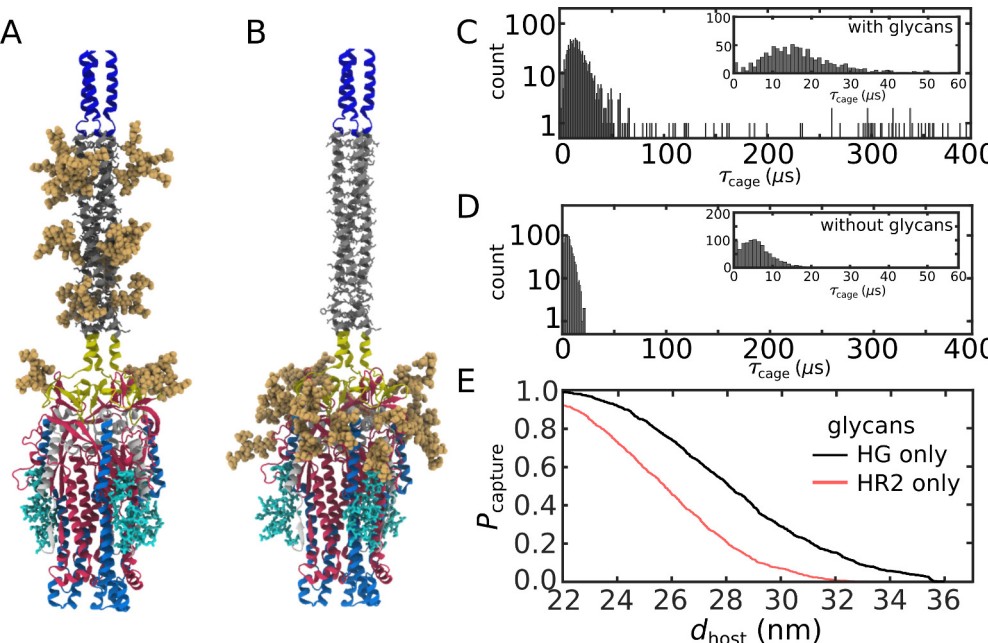

**Appendix 1—figure 6.** Relative influence of glycans on HR2 and HG. (**A**) Structural model with only glycans shown on HG. (**B**) Structural model with only HR2 glycans present. (**C**) Distribution of timescales with only HG glycans present. (**D**) Distribution with only HR2 glycans present. (**E**) Probability that $d_{FP} > d_{host}$ for at least one FP. There is a higher probability of extending to $d_{host}$ when only HG glycans are present than when only HR2 glycans are present (black vs. red curves).

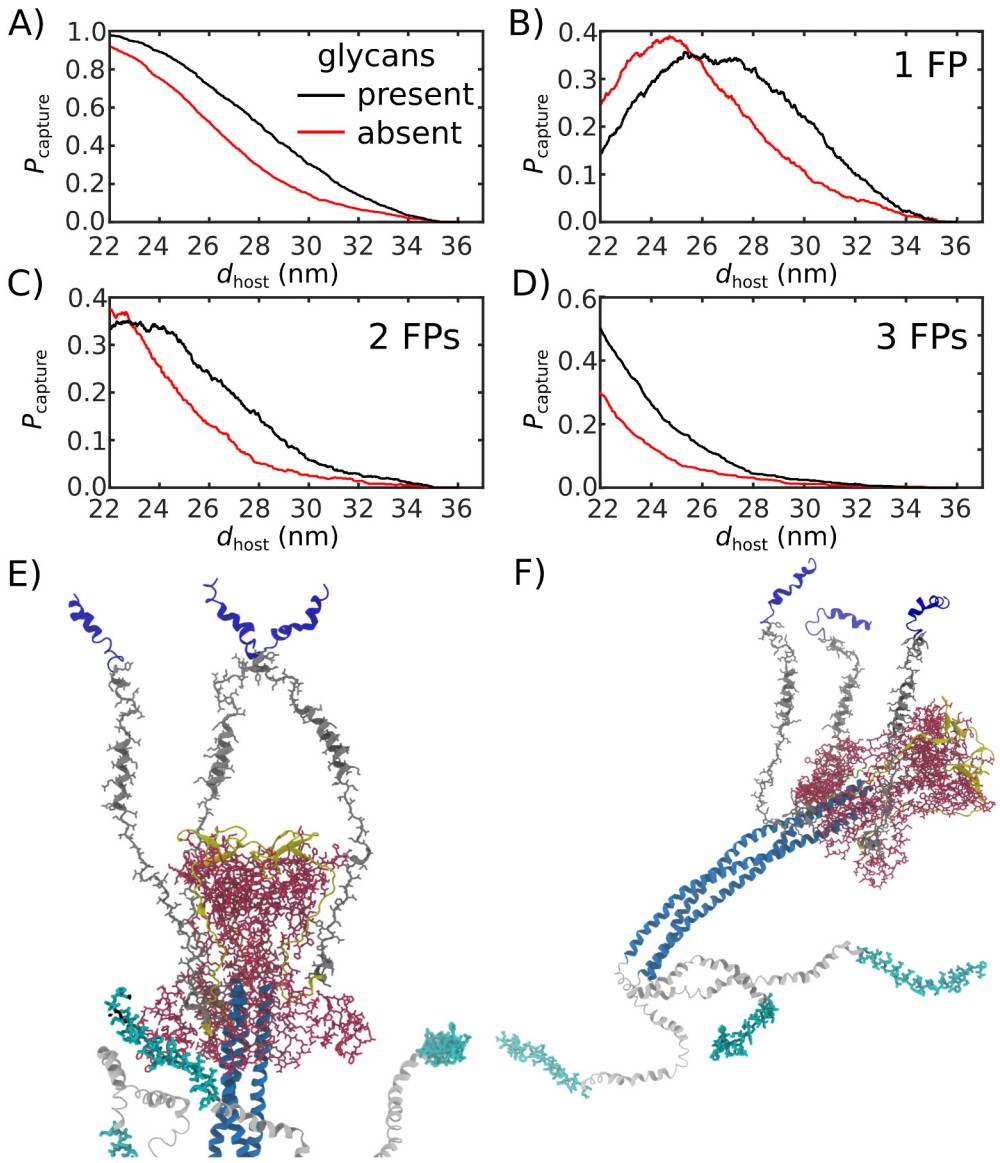

**Appendix 1—figure 7.** Glycans promote host capture with dissociated TM region. (**A-D**) Even in the case where the TM strands are able to dissociate, the presence of the glycans increases the probability that the FPs will capture the host membrane. 1000 transitions were simulated for each system. (**E–F**) Snapshots of glycosylated Spike protein where the TM strands dissociate from one another (glycans are not shown).

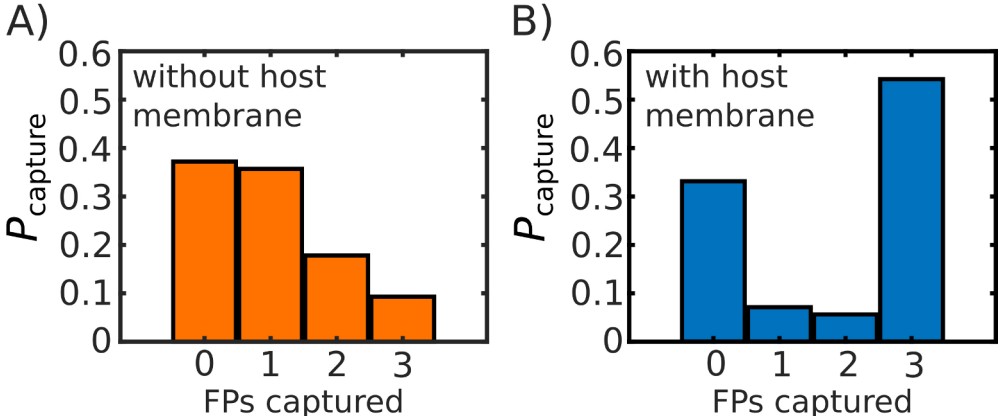

**Appendix 1—figure 8.** Comparison of probability of FP capture with, and without, a host membrane potential. Probability of capture, calculated from 1000 simulated transitions. (**A**) Model with no host membrane, using $d_{host} = 27$ to define capture. (**B**) Model with host membrane potential as defined in *Equation S1*, which traps (i.e. potential begins to decrease) FPs at ~27 nm from the virtual viral membrane. Both (A) and (B) show approximately the same probability of 0 FPs being captured, while (B) shows a drastic increase on the probability that capture will involve all three FPs.

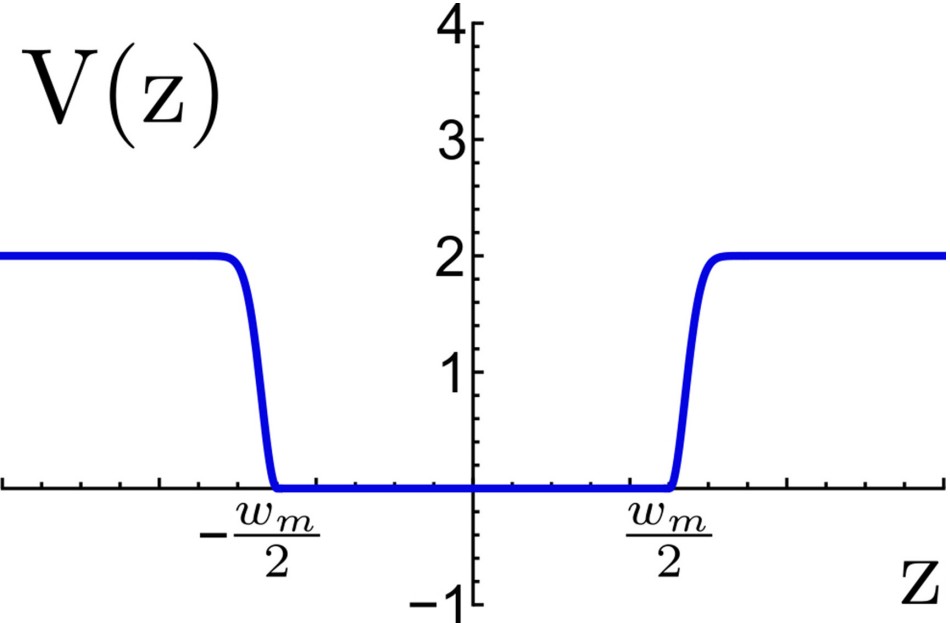

**Appendix 1—figure 9.** Effective potential for TM confinement in a virtual viral membrane. The flat-bottom region represents the virtual membrane of width $w_m = 5$ nm, this region allows the TM motif to move freely between the planes $z = \frac{w_m}{2}$ and $z = \frac{-w_m}{2}$, beyond the flat region an energetic penalty is included to restrain the TM to escape the virtual membrane. Energy is measured in reduced energy units.

