## [Decision Letter]

**Acceptance summary:**

This paper describes a vital step in the infection cycle of the Covid-19 virus, providing useful insights into how the virus enters the cell and the importance of the modification to viral proteins with glycans. Despite the use of a simplified model to describe the system, this study provides a thorough examination of the conformational changes of the Covid-19 Spike protein, knowledge that could be exploited for drug design purposes and would otherwise have been impossible to obtain with a more detailed model.

**Decision letter after peer review:**

Thank you for submitting your article "Sterically-Connned Rearrangements of SARS-CoV-2 Spike Protein Control Cell Invasion" for consideration by *eLife*. Your article has been reviewed by 3 peer reviewers, and the evaluation has been overseen by a Reviewing Editor and a Senior Editor. The following individual involved in review of your submission has agreed to reveal their identity: Kei-ichi OKAZAKI (Reviewer #2).

Essential revisions:

1) For data reproducibility, both pre-fusion and post-fusion structure models should be provided as coordinate files in the SI files. Based on limited available information, authors made complete structure models, which itself has considerable values for subsequent studies.

(1a) Related to this, it is not explicitly stated that the structure drawn in Figure 1C is the post-fusion model the authors used in this work. If not, the post-fusion structure must be provided for example alongside Figure S1.

(1b) In the structure in Figure 1C, 3 TMs look asymmetric. Since TMs are not included in the original PDB model, authors modeled it. How this particular asymmetric configuration appeared?

(2) In the simulation, TMs are restrained to the implicitly-assumed membrane plane, which is reasonable. However, in the snapshots in Figures 2 and 4, the tilt angle of TMs does not seem to be restrained. Is it possible that this slant TMs may affect the overall orientation of the spike proteins and thus could affect the d_FP_? Can author comment on it?

(3) In the analysis of Figure 5, once one FP captures the host membrane, this must anchor the FP subsequently. This anchoring must affect the dynamics of the rest of spike protein including the other FPs. Therefore, in reality, there must have higher probabilities of multiple FP captures (2FP and 3FP). Can authors conduct some simple simulations in which FP is anchored once it reaches the host cell membrane.

(4) In the description of all-atom structure-based model, The meanings of "1-3 dihedral potentials" and "6-12 interactions" are not clearly understood.

(5) It was difficult to understand the viral membrane potential described in Equation S1. What does "*" mean in the upper equation and does K take a positive or negative value? It might help to include a plot of the potential along z.

(6) It would be useful to have a detailed Discussion section on caveats of the model/potential alternate mechanisms as well as possible predictions which can be tested which include the following facts/questions. The alternate mechanisms need not be simulated. They can be speculations based on the present model. The indicated citations are just suggestions and may not represent the latest results. The authors should check for those.

(6a) The starting prefusion structure for the simulations is a hypothetical structure of what the spike would look like post cleavage. This should be emphasized. Does the frustration analysis indicate that some regions will already be unfolded pre-cleavage? For instance HR2 could already be unfolded before cleavage and then will it be able to partially cage some part of the spike? And what would this do to the mechanism?

(6b) What changes in mechanism could occur if this were a dual structure-based model instead of a single structure-based one? Do any of these seem physically reasonable?

(6c) Does the importance of glycosylation decrease if HR1 folds faster than the timescale of uncaging of HG without the glycosylation? What increase in contact (or dihedral) strengths or contact to dihedral ratios would be required for this to happen? Are these increases physically reasonable?

(6d) The TM helices are pinned into a trimeric conformation. What would happen if they are not and can move away from each other? See for instance: https://www.biorxiv.org/content/10.1101/2021.06.07.447334v1

(6e) Some percentage of spikes are naturally in the post-fusion conformation. See for instance: https://science.sciencemag.org/content/369/6511/1586 How does that fit into the simulation results?

(6f) Please look through current literature (since this is a fast moving field) for the role of spike sugars in fusion. See for instance https://elifesciences.org/articles/61552 Are there experimental results which support the present model or can the authors suggest experiments which can test the model specifically in the context of sugar placement?

(7) Throughout: Viral membrane envelopes are not called "capsids".

(8) Lines 52-57: These sentences imply an order to the cleavage/ACE2 binding events (ACE2 binding happens after cleavage). Has this been proven? If yes, please give a reference. If not, please reword.

(9) Figure 1: Please also give PDB IDs for the structures right here.

(10) Lines 153 onwards: Since at least some model justification depends on frustrated contacts, it would be useful to explain frustrated contacts in some detail and bring the supplementary figure into the main paper (if no page limit constraints exist).

(11) Figure 5B: Please state if this figure is for a glycosylated protein.

(12) Line 290: Please remove the word dramatic. It's not a quantifiable amount.

(13) Lines 301-305: I don't understand what happens when FPs transition to a post-fusion conformation without engaging the host membrane. What does this mean in the context of the present model? And what happens in hemagglutinin? It would be useful to have a clarification of these sentences and a detailed explanation.

*Reviewer #1:*

Authors performed all-atom structure-based molecular dynamics simulation of the conformational change process of SARS-CoV-2 spike protein from its pre-fusion form to the post-fusion form, with and without glycans bound to the spike protein. Authors found that the bound glycans provide considerable steric barrier in the transition, which prolongs the transition compared to the case without the bound glycan. Interestingly, this intermediate configuration, called caged state, has high probability to extend its fusion peptide towards the host cell membrane. Thus the bound glycan enhances the probability for the spike protein to capture the host cell membrane.

Using a simplified energy function, i.e. the structure-based model, authors could simulate extremely large-scale conformational change numerous times, which robustly shows the role of the bound glycans., which strengthens their finding.

On the other hand, any attraction interactions of glycans with protein amino acids are completely ignored in this simplified energy function, which is a clear limitation of the current study. These interaction, if included, could give much longer pause in the caged intermediate. Thus, real effect of the bound glycans may be even stronger.

*Reviewer #2:*

Dodero-Rojas et al. investigated a large-scale conformational change of the SARS-Cov-2 Spike protein involved in membrane fusion to its host cell. They used a structure-based all-atom model to simulate the large-scale conformational transition between prefusion and post-fusion conformations, which is impossible to simulate with conventional simulation methods. From extensive simulations, they identified the "caged" intermediate, which is realized through steric interactions of glycans. It was clearly shown from various control simulations that the caged intermediate has a much shorter lifetime without glycans, and glycans attached to the head group (HG) are mainly responsible for the stable intermediate. Furthermore, they showed that the caged intermediate facilitates capturing of the host membrane by extending the fusion peptides (FP) in the perpendicular direction to the membrane. The probability of capturing the host membrane by FP is higher with the stable caged intermediate in the presence of glycans for the virus-host inter-membrane distance of 30 nm, consistent with the cryotomography observations. Overall, the author's claims and conclusions are justified by their data. The strength of this work is that they simulated the unprecedently large conformational transition thousands of times. The weakness is that they used a simplified model that might miss physical interactions like electrostatic interactions. However, this work can establish a foundation for more detailed simulations with precise physicochemical interactions.

*Reviewer #3:*

The conformational transition of the SARS-CoV2 spike protein from its prefusion state to its post-fusion state helps the viral envelope fuse with the host membrane and eject the viral RNA into the host cell. This conformational transition is difficult to study in its entirety using either experimental techniques or standard simulation methods. Here, the authors use starting structures modelled on the prefusion structure and a potential energy function encoding the post-fusion structure to characterize the spike conformational transition using molecular dynamics (MD) simulations. Such structure-based models simplify the potential energy function and are able to simulate timescales many orders of magnitude longer than those seen in standard atomistic MD simulations. The spike simulations show that a helical region (HR2) C-terminal to the spike head (HG) which links HG to the transmembrane segment (TM; embedded in the viral membrane) unfolds and cages HG close to the viral membrane. Glycosylation increases the size and roughens the surface of HG allowing it to be caged for longer by HR2. This allows the helical region (HR1) N-terminal to HG to fold into a long helical trimer and potentially reach out to and latch onto the host membrane more effectively suggesting that glycosylation enables efficient fusion.

Structure-based models encode the protein structure in the potential energy function simplifying it. They also do not encode interactions which are not present in the structure. In the model used here, this implies that not all interactions present in the starting prefusion structure are stabilizing. The viral membrane is encoded implicitly while the host membrane is excluded and the sugars are included using connected beads which incorporate the sugar structure but ignore any further attractive interactions. However, such simplified models have a grounding in protein folding theory. Additionally, it is such simplifications which allow the model to simulate the spike conformational transition. Finally, similar models have previously been successfully used to understand the conformational transitions of large molecular machines such as the ribosome.

The authors have successfully simulated the spike conformational transition, shown a likely order of events that occur during this conformational transition and illustrated the potential importance of spike glycosylation to SARS-CoV-2 fusion.

---

## [Author Response]

Essential revisions:1) For data reproducibility, both pre-fusion and post-fusion structure models should be provided as coordinate files in the SI files. Based on limited available information, authors made complete structure models, which itself has considerable values for subsequent studies.

We are pleased that our structural models can be of utility to other groups. Accordingly, as part of the SI, we now provide PDB files of the structural models used in the study.

(1a) Related to this, it is not explicitly stated that the structure drawn in Figure 1C is the post-fusion model the authors used in this work. If not, the post-fusion structure must be provided for example alongside Figure S1.

Thank you for noting this ambiguity in the text. To clarify this point, we have introduced two modifications.

– The last paragraph of the introduction now specifies that the post-fusion structure used to define the force field is shown in Figure 1C.

“Simulations were initiated with the Spike protein in the prefusion conformation, while the energetics were defined to favor the post-fusion conformation (shown in Figure 1C)”

– As recommended, we also now show this structure alongside the pre-fusion structure in Figure S1.

(1b) In the structure in Figure 1C, 3 TMs look asymmetric. Since TMs are not included in the original PDB model, authors modeled it. How this particular asymmetric configuration appeared?

The TM region for the post-fusion structure was modelled as alpha-helices without enforcing a symmetry constraint. However, since the energetics of the TM were defined entirely by the pre-fusion conformation, the post-fusion representation of the TM is only intended to depict the putative position of the TM after fusion occurs. To clarify this point, in the Methods and Materials section “Structural Modeling of the Spike Protein” we now explicitly state that, for the post-fusion model, the CR and FP regions were modelled with symmetry restraints imposed, while the TM was modelled as helices without symmetry restraints. Page 12 reads:

“For the post-fusion structure, unresolved residues in FP and CR were modeled as helical regions, using the automodel module of Modeller 9.24 with symmetry restraints imposed. […] As a note, the post-fusion configuration of the TM region was not used to define any aspect of the structure-based model.”

(2) In the simulation, TMs are restrained to the implicitly-assumed membrane plane, which is reasonable. However, in the snapshots in Figures 2 and 4, the tilt angle of TMs does not seem to be restrained. Is it possible that this slant TMs may affect the overall orientation of the spike proteins and thus could affect the d_FP_? Can author comment on it?

Thank you for raising this interesting point. As noted, the TM motif is restrained to a plane without an orientation bias. To determine whether tilting of the TM region has a significant impact on the reach of the FPs, we performed additional analysis of the simulations and found that there is no distinguishable difference in orientations within the two sets (glycans present, or absent) of simulations. This comparison is now presented in the SI Results section entitled “TM tilt angle distributions” (page 1), which includes a new SI Figure (Figure S2). As described in the SI, the glycosylated and degycosylated Spike proteins exhibit similar ranges of TM tilting, which is consistent with the reach of each FPs being equally affected by the TM tilt. In addition, the tilt angle formed with the plane normal rarely exceeds 45°. Accordingly, this indicates that the differential reach of the glycosylated and unglycosylated systems will be robust to possible tilting of the TM in the simulations.

To address this point, we have also added the following passage to the main text (page 8):

“To ask whether this differential extension of the FP is due to tilting of TM, we calculated the distribution of angles between the TM bundle and the viral membrane (Figure S2). This revealed that the TM tilting distributions were similar for the two systems, which indicates that the differential dynamics of the FPs can not be attributed to this aspect of the model..”

(3) In the analysis of Figure 5, once one FP captures the host membrane, this must anchor the FP subsequently. This anchoring must affect the dynamics of the rest of spike protein including the other FPs. Therefore, in reality, there must have higher probabilities of multiple FP captures (2FP and 3FP). Can authors conduct some simple simulations in which FP is anchored once it reaches the host cell membrane.

This is a very interesting point. It is indeed expected that the anchoring of one FP will affect the probability of the other FPs binding to the host membrane. In order to explore this relationship, we extended our model, such that each FP can irreversibly bind the host mem brane. This was enabled by introducing a second flat-bottom potential (Equation S1) to describe the host membrane region. We then used this model to perform 1000 simulations of the glycosylated Spike protein. As expected, the probability that at least one FP reaches the host membrane is not affected by the introduction of the host membrane potential (Figure S8). However, as suggested by the reviewers, there is a significant increase in the probability that all three FPs reach the host membrane region. These results are now described, in detail, in a new SI section entitled “Effect of a virtual host membrane on FP capture probabilities”. We have also added a new paragraph to the main text that describes these additional calculations (page 10).

“In terms of the mechanistic features of membrane fusion, one can expect that FP binding to a host membrane will impact the probability that subsequent FPs will also bind. […] Together, this analysis suggests that the primary mode for Spike protein mediated membrane fusion is through use of a “sequential” mechanism.”

(4) In the description of all-atom structure-based model, The meanings of "1-3 dihedral potentials" and "6-12 interactions" are not clearly understood.

Thank you for pointing this ambiguous notation. We expanded the description of the model, and included the full form of the structure-based model potential on page 2 of the SI.

(5) It was difficult to understand the viral membrane potential described in Equation S1. What does "*" mean in the upper equation and does K take a positive or negative value? It might help to include a plot of the potential along z.

Thank you for nothing this typesetting issue. We now include a corrected version of the membrane potential on page 2 of the SI. Also, following the reviewers’ suggestions we included a plot of the potential (Figure S9).

(6) It would be useful to have a detailed Discussion section on caveats of the model/potential alternate mechanisms as well as possible predictions which can be tested which include the following facts/questions. The alternate mechanisms need not be simulated. They can be speculations based on the present model. The indicated citations are just suggestions and may not represent the latest results. The authors should check for those.

Thank you for raising this suggestion. Accordingly, we now include an extended discussion of the caveats of the model (page 11). We also expanded the discussion by including experimental strategies that could be used to explore the effect of Spike glycosylation on the membrane fusion process (page 11), and thereby test the predictions of the current study.

“The current predictions suggest that the steric composition of the Spike protein and glycans can guide the global dynamics of host-membrane capture. […] If sterics dominate the dynamics, as predicted by the structure-based model, then mutations to HG glycan sites should significantly reduce the probability of membrane capture.”

(6a) The starting prefusion structure for the simulations is a hypothetical structure of what the spike would look like post cleavage. This should be emphasized.

We included in the last paragraph of the introduction (page 4) that the pre-fusion structure is a possible snapshot of the system after cleavage and S1 dissociation.

“It is important to emphasize that the prefusion model used as a starting point is intended to represent a state in which S2’ cleavage and S1 dissociation have occurred. While the precise timing of these steps is unknown, we assume they can occur prior to any significant conformational changes in the S2 subunit.”.

Does the frustration analysis indicate that some regions will already be unfolded pre-cleavage? For instance HR2 could already be unfolded before cleavage and then will it be able to partially cage some part of the spike? And what would this do to the mechanism?

The frustration analysis shows that the most frustrated region of HR2 corresponds to residues near the TM boundary. Since the frustration analysis is based on protein sequences in solution, one explanation for this localized frustration is that this region is only marginally stable in solution, and likely engages in stabilizing interactions with the viral membrane. In support of this interpretation, tomographic imaging of HR2 has revealed the presence of two distinct kinks, called the ”ankle” and ”knee” [1], while the remainder is described well by a coiled coil motif.

(6b) What changes in mechanism could occur if this were a dual structure-based model instead of a single structure-based one? Do any of these seem physically reasonable?

This is a very interesting question. If a multi-basin model were implemented, where the prefusion conformation were stabilized, we would expect several substeps of the rearrangement to slow down. As described in the earlier reply (Comment 6), we have added discussion on these possibilities in the main text.

(6c) Does the importance of glycosylation decrease if HR1 folds faster than the timescale of uncaging of HG without the glycosylation? What increase in contact (or dihedral) strengths or contact to dihedral ratios would be required for this to happen? Are these increases physically reasonable?

Thank you for raising this interesting point. In our “downhill” model, we observe that the HR1 folds rapidly, where there are no obvious large-scale barriers. In the absence of an apparent free-energy barrier for the formation of HR1 (inferred by probing the number of contacts formed), increasing the slope of the downhill potential will have minimal effects on the mean- first passage time. While one could artificially strengthen the HR1 post contacts, doing so would also lead to a hyperstable helical bundle, where folding temperatures would be non-physical (e.g. 600K). Accordingly, one could only argue that the HR1 contact strengths should be increased by 10-20% (at most), which would only affect the HR1 formation time by a few percent (assuming diffusive dynamics [2] on a downhill 1D free-energy surface). Since the current model may be described as a “fast-forming HR1” model, it is more likely that the predicted impact of glycosylation will be more pronounced in solution.

(6d) The TM helices are pinned into a trimeric conformation. What would happen if they are not and can move away from each other? See for instance: https://www.biorxiv.org/content/10.1101/2021.06.07.447334v1

To address this question, we generated another variant of our model, where the harmonic intra-TM interactions were replaced with marginally-stable 6-12 Lennard-Jones interactions. With this model, we simulated 1000 transitions for the glycan-present and glycan-absent systems (2000 events, in total). We find that, even if the TM strands dissociate from one another, the FPs reach further in the glycosylated system. These results are shown in a new SI figure (Figure S7). We also added the following text to the main manuscript (page 8).

“To assess whether the influence of glycans on FP dynamics is robust, we considered a variant of our model in which the TM helices may dissociate from each other. […] These simulations show that the differential dynamics of the FPs is robust to the precise description of the TM bundle.”

(6e) Some percentage of spikes are naturally in the post-fusion conformation. See for instance: https://science.sciencemag.org/content/369/6511/1586 How does that fit into the simulation results?

In Cai et al. 2020, the post-fusion structures appear in virons under mild detergent conditions, where spontaneous activation can occur without host-virus interactions. In our study, we consider activation upon host-virus interaction. Accordingly, the Spike proteins that fail to reach the host boundary in our simulations describe a subpopulation of Spike protein that do not spontaneously adopt a post-fusion state in the absence of a host. Page 8 now reads:

“Finally, while the Spike Protein has been experimentally observed to spontaneously transition from the prefusion to post-fusion configuration (Cai et al. 2020), the probabilities reported above describe events that occur when the Spike Protein is activated through host-virus interactions.”

(6f) Please look through current literature (since this is a fast moving field) for the role of spike sugars in fusion.

As recommended by the reviewers, we went through the newest literature. Some of these new papers served as foundation to address multiple revisions. As an example, the interaction of the fusion peptides with the host membrane has been studied in [3], which allowed us to justify our host membrane potential used to answer previous points.

See for instance https://elifesciences.org/articles/61552.

We are glad that the reviewers have brought this paper to our attention. Its results fit well with our findings, as the inhibition of glycan elaboration reduced the viral entry drastically. We expanded the Results section to include this supportive experimental finding. Page 8 now reads:

“Therefore, these simulations suggest the probability of infection would drop substantially if the Spike was not glycosylated. Consistent with this, experimental measurements have found that inhibiting the production of glycans decreases the efficiency of host cell entry (Yang et al., 2020).”

Are there experimental results which support the present model or can the authors suggest experiments which can test the model specifically in the context of sugar placement?

Thank you for this suggestion. As described for Comment 6 (above), we included a section with possible experimental tests that could further quantify the influence of glycosylation on the Spike protein rearrangement.

(7) Throughout: Viral membrane envelopes are not called "capsids".

Thank you for raising the distinction, we corrected the wording issue.

(8) Lines 52-57: These sentences imply an order to the cleavage/ACE2 binding events (ACE2 binding happens after cleavage). Has this been proven? If yes, please give a reference. If not, please reword.

To the best of our knowledge, the order of events has not been unambiguously determined experimentally. For our purposes, we simply assume they both complete prior to any large-scale deformations of S2. To clarify this point, we have rephrased to avoid confusion. Page 2 now reads:

“While the order the S2’ cleavage and S1/S2 dissociation is not known, it is generally thought that both processes occur prior to any large-scale rearrangements of S2.”

(9) Figure 1: Please also give PDB IDs for the structures right here.

The PDB IDs are now included to the caption of Figure 1.

(10) Lines 153 onwards: Since at least some model justification depends on frustrated contacts, it would be useful to explain frustrated contacts in some detail and bring the supplementary figure into the main paper (if no page limit constraints exist).

Thank you for raising this point. While the results of frustration analysis support the use of a structure-based model, the original text gave the impression that the model was defined based on frustration analysis. To make it clear that we did not introduce frustrated contacts, we have significantly revised the passage in question (pages 4-6).

“Before describing the simulated events, it is valuable to discuss the analysis of energetic frustration(Ferreiro et al., 2007;Parra et al., 2016) in the Spike protein, which supports the application of an unfrustrated model. […] Accordingly, any frustration in these regions that was not included in the model is not likely to influence the primary finding of the current study.”

(11) Figure 5B: Please state if this figure is for a glycosylated protein.

Thank you for pointing this out. It is a glycosylated protein. We have updated the caption to reflect this point.

(12) Line 290: Please remove the word dramatic. It's not a quantifiable amount.

We removed the word ”dramatic”.

(13) Lines 301-305: I don't understand what happens when FPs transition to a post-fusion conformation without engaging the host membrane. What does this mean in the context of the present model? And what happens in hemagglutinin? It would be useful to have a clarification of these sentences and a detailed explanation.

We thank you for pointing this confusing phrasing. This part of the main text is now rephrased to clarify the two possible modes of capture found in fusion protein class I, such as hemagglutinin. The first mode is ”cooperative” where less than 3 FPs are attached to the host membrane. The ”sequential” mode refers to the case where all 3 FPs reach and capture the host membrane. To clarify this, we have replaced the text on page 9 with:

“When glycans are absent, there is a marginal probability that only one or two FPs will reach the membrane (Figure 5D-E), where the other FPs would likely transition directly to their post-fusion orientations without engaging the host. […] When all 3 FPs attach to the host membrane, the dynamics may be described in terms of the so-called “sequential” mechanism of fusion (Lin et al., 2014).”

References:

1. B. Turonova, M. Sikora, C. Schurmann, W. J. Hagen, S. Welsch, F. E. Blanc, S. von Bulow, M. Gecht, K. Bagola, C. Horner, et al., Science 370, 203 (2020).

2. J. D. Bryngelson and P. G. Wolynes, The Journal of Physical Chemistry 93, 6902 (1989).

3. D. Gorgun, M. Lihan, K. Kapoor, and E. Tajkhorshid, Biophysical Journal (2021).